# Comparison of the imaginary parts of the atmospheric refractive index structure parameter and aerosol flux based on different measurement methods

Renmin Yuan[a] Hongsheng Zhang[b] Jiajia Hua[c] Hao Liu[d] Peizhe Wu[a] Xingyu Zhu[a] Jianning Sun[e]

[a] School of Earth and Space Science, University of Science and Technology of China, Hefei 230026, PR China
[b] Laboratory for Climate and Ocean-Atmosphere Studies, Department of Atmospheric and Oceanic Sciences, School of Physics, Peking University, Beijing 100081, PR China
[c] China Meteorological Administration Xiong'an Atmospheric Boundary Layer Key Laboratory, Xiong'an New Area 071800, PR China
[d] School of Mathematics and Physics Anhui Jianzhu University, Hefei 230601,PR China
[e] School of Atmospheric Sciences, Nanjing University, Jiangsu, 210093, PR China

Correspondence: Yuan Renmin (rmyuan@ustc.edu.cn)

Abstract: The complexity of aerosol particle properties and the diversity of characterizations make aerosol vertical transport flux measurements and analysis difficult. Although there are different methods, such as aerosol particle number concentration flux and aerosol mass flux based on the eddy covariance principle, and aerosol mass flux measurements based on the light-propagated large-aperture scintillation principle, there is a lack of mutual validation among the different methods. In this paper, a comparison of aerosol mass flux measurements based on the eddy covariance principle and aerosol mass flux measurements based on the light-propagated large aperture scintillation principle is carried out. The key idea of aerosol mass flux measurements based on the light-propagated large-aperture scintillation principle is the determination of the imaginary part of the atmospheric equivalent refractive index structure parameter (AERISP). In this paper, we first compare the AERISPs measured by two different methods and then compare the aerosol mass vertical transport fluxes obtained by different methods. The experiments were conducted on the campus of the University of Science and Technology of China (USTC). A light propagation experiment was carried out between two tall buildings to obtain the imaginary and real parts of the AERISPs for the whole path, which in turn can be used to obtain the aerosol vertical transport flux. An updated visibility meter is installed on the meteorological tower in the middle of the light path, which is utilized to obtain the single-point visibility, which is then converted to the imaginary part of the atmospheric equivalent refractive index (AERI). The imaginary parts of the AERISP were obtained via spectral analysis with visibility data. The results show that the imaginary parts of the AERISPs obtained by different methods are in good agreement. The imaginary part of the AERI measured by the visibility meter and the vertical wind speed obtained by the ultrasonic anemometer were used for covariance calculations to obtain the aerosol vertical transport flux. The trends in aerosol vertical transport fluxes obtained by the different methods are consistent, and there are differences in some details, which may be caused by the inhomogeneity of the vertical transport of aerosol fluxes. The experimental results also showed that urban green land is a sink area for aerosol particles.

Keywords: light propagation, scintillation, atmospheric equivalent refractive index, structure parameter, eddy covariance, aerosol fluxes

# 1 Introduction

Atmospheric aerosols are solid or liquid particles suspended in the atmosphere that can affect public health, reduce near-surface visibility, decrease direct radiation from the air, and act as condensation nuclei affecting cloud structure and distribution(McNeill, 2017;Rosenfeld et al., 2014). Human activities have dramatically altered air quality, climate and the Earth system. The expansion of urban, agricultural and industrial areas and changes in the nature of land use have increased aerosol concentrations. Due to the complexity of aerosols, many observations have been carried out from different perspectives. However, most of the current observations only measure the state characteristics of aerosols, such as concentration, particle size distribution, and composition, and what is obtained is an average characterization of aerosol properties(Krieger et al., 2012).

Aerosol particles in the atmosphere follow atmospheric motion, which manifests as an uneven distribution of aerosol particle concentrations in space and time. On the one hand, unevenly distributed aerosol particles will have a corresponding effect on light wave propagation in the atmosphere, on the other hand, we can understand the distribution characteristics of aerosol particles based on the optical effect of aerosol particles and then obtain more information about the transportation of aerosols.

Previously, Yuan et al. (2016) introduced the concepts of the atmospheric equivalent refractive index (AERI) and the atmospheric equivalent refractive index structure parameter (AERISP). The term AERISP corresponds to the equivalent medium containing air and aerosol particles, relative to the commonly used atmospheric refractive index structure parameter (RISP) obtained from single-point measurements (Wyngaard et al., 1971). The AERI includes real and imaginary parts; accordingly, the AERISP also includes real and imaginary parts of the structure parameters. When the working wavelength is in the atmospheric transparent band, the light wave is almost not absorbed by the gas components in the atmosphere, and the attenuation of the light wave is caused mainly by the extinction of aerosol particles. Theoretical analysis revealed that, the imaginary part of the AERISP is determined by the fluctuation in aerosol concentration, and the real part of the AERISP corresponds to the atmospheric temperature variation (Yuan et al., 2021). Experiments have shown that aerosol particles follow the same theory of locally homogeneous isotropic turbulence as air molecules(Martensson et al., 2006;Vogt et al., 2011a;Ren et al., 2020); that is, the fluctuation in the particle concentration follows the '-5/3' power law under unstable atmospheric stratification, and the concentration-velocity cospectra for particle number flux follow the '-4/3' power law, similar to the temperature-velocity cospectra (Kaimal et al., 1972). Therefore, the distribution of small particles can be regarded as a passive conservative quantity, similar to the temperature field.

Then, it can be assumed that the aerosol mass concentration fluctuation also follows the locally
homogeneous isotropic turbulence theory; thus the aerosol mass concentration structure parameter
can be defined (Yuan et al., 2016). Based on the fact that the temperature structure function satisfies
the surface layer similarity theory and thus the surface layer sensible heat flux is obtained from the
temperature structure parameter (Wyngaard et al., 1971), it is analogous to that, based on the fact
that the aerosol mass concentration structure parameter satisfies the surface similarity theory, the
surface layer aerosol mass flux is obtained from the aerosol mass concentration structure parameter
(Yuan et al., 2016;Yuan et al., 2019). Using the relationship between the temperature-real part of the
AERI and aerosol mass concentration-imaginary part of the AERI, the temperature structure
parameter and aerosol mass concentration structure parameter are obtained from the real part of the
AERISP and imaginary part of the AERISP.
From this, the aerosol mass concentration flux can be obtained, utilizing large aperture
scintillometer (LAS) measurements. AERISP observations are carried out in many places, after
which the aerosol flux is obtained via the similarity theory (Yuan et al., 2016;Yuan et al., 2019).
However, there is a lack of experimental verification of the imaginary structure parameter and
aerosol flux observations. Currently, the imaginary part of the AERISPs is obtained using only LAS
measurements; therefore, it is necessary to carry out measurements of the imaginary part of the
AERISPs based on other methods, as well as measurements of aerosol fluxes based on different
methods.
At present, in addition to the previously mentioned measurements of the AERISP based on the
principle of long-range light propagation and the similarity theory of the surface layer to obtain the
vertical transport flux of aerosol mass in the surface layer, several studies utilize eddy covariance
(EC) techniques with fluctuations in aerosol particle number concentration and fluctuation in vertical
wind speed to obtain the flux of the number concentration of aerosol particles. Such measurements
are carried out in many places (Gordon et al., 2011;Vogt et al., 2011b;Ripamonti et al., 2013).
Measurements have revealed quantitative relationships for urban aerosol fluxes among urban vehicle
emissions and meteorological conditions (Jarvi et al., 2009), and characteristics of sea salt transport,
and aerosol properties (Nemitz et al., 2009). We followed this approach and conducted several
measurements in 2019 and 2020 to measure aerosol particle number concentration fluxes using an
eddy-correlation system combining a fast-response particle counter with an ultrasonic anemometer
with a response frequency of up to 10 Hz, and to calculate aerosol mass concentration fluxes by
simultaneously measuring aerosol particle size distribution, mass concentrations, forward scattering
coefficients, extinction coefficients, and other parameters. For half of the experimental period, the
trends of the measurements of the two methods were the same, while the other periods differed more
(unpublished experimental results). The main reason may be the very weak extinction of aerosol
particles at scales much smaller than the working wavelength. Second, the aerosol number
concentration flux must be combined with parameters such as particle size distribution, complex
refractive index of aerosol particles and aerosol particle density, which also need to be sampled in
real time. This also illustrates the complexity of aerosol particles.
Recently, Ren et al. (2020) improved upon the conventional visibility meter method to obtain 1
Hz visibility data, and subsequently utilized the EC method to obtain the aerosol vertical transport
flux based on the relationship between visibility and aerosol mass concentration. The visibility is
approximately inversely proportional to the atmospheric extinction coefficient, i.e., approximately
inversely proportional to the imaginary part of the AERI; therefore their theory of obtaining the
aerosol vertical transport flux by the EC method is close to theory of the aerosol vertical transport
flux based on the measurement of the long-path light propagation. Inspired by their work, we used
an improved visibility meter in this study to obtain visibility data at a higher frequency of 1 Hz and
cross-correlated the data with ultrasonic anemometer measurements to potentially utilize the
obtained aerosol vertical transport fluxes to achieve experimental validation of the imaginary part of
the AERISP and aerosol flux observations.
The theoretical and experimental introduction is given in the second part of the paper, the
experimental results are given in the third part, and the conclusions and discussion are given in the
fourth part.

# 2 Theoretical methods and experiments

The AERISP and aerosol vertical transport flux are the topics of interest in this paper. In this
section, definitions and theoretical expressions for these parameters are given, as well as how the
measurements were carried out.

## 2.1 The imaginary part of the AERISP

Normally, the atmosphere consists of gas molecules and aerosol particles. When a light beam
propagates in the atmosphere, due to the inhomogeneous distribution of the atmospheric gas
refractive index, the beam will be refracted and diffracted, which results in an inhomogeneous spatial
distribution of the beam energy. Due to the existence of aerosol particles in the atmosphere, the beam
will be scattered and absorbed, and the energy of the beam will be weakened. Therefore, the
atmospheric gas molecules and aerosol particles can be taken as a whole, called the equivalent
medium; thus, the atmospheric equivalent refractive index (AERI) $n_{equ}$ concept is introduced (van
de Hulst, 1957;Yuan et al., 2021),

$$n_{equ} = n_m + i\frac{2\pi}{\eta^3}\int_0^\infty S(0)\frac{dN}{dD}dD \qquad (1)$$

where $n_m$ is the refractive index of atmospheric molecules, $\eta$ is the wavenumber of light waves, and
$i$ denotes an imaginary number. S(0) is the forward scattering function (0 in parentheses is the
scattering angle). $N$ is the number of aerosol particles per unit volume, and $dN/dD$ is the size
distribution of aerosol particles.
The AERI consists of real and imaginary parts denoted by $n_{Re}$ and $n_{Im}$, respectively; i.e., $n_{equ}$
$=n_{Re} +in_{Im}$. The real part is the refractive index of the molecule, and the imaginary part is.
$$n_{Im} = \frac{2\pi}{\eta^3} \int_0^\infty Re[S(0)] \frac{dN}{dD} dD \tag{2}$$
The atmospheric extinction coefficient has a similar form(Liou, 2002):
$$\beta_{ext} = \frac{4\pi}{\eta^2} \int_0^\infty Re[S(0)] \frac{dN}{dD} dD \tag{3}$$
From Eqs (2) and (3), we can see that
$$n_{Im} = \lambda\beta_{ext}/4\pi \tag{4}$$
where $\lambda$ is the working wavelength ($\lambda=2\pi/\eta$).
Due to the dependence of the reduction in contrast on atmospheric absorption and scattering,
the following relationship between visibility $V$ and extinction coefficient $\beta_{ext}$ can be obtained:
V=3.912/$\beta_{ext}$ (Middleton, 1957;Charlson, 1969). Thus, $\beta_{ext}$ in the relationship (V=3.912/$\beta_{ext}$)
represents the extinction by all compositions in the air, e.g., absorption and scattering of aerosols
and atmospheric molecular extinction. In other words, the visibility-based extinction coefficient is
sum of the extinction coefficient from aerosol absorption and scattering and the atmospheric
molecular extinction coefficient. However, in the urban atmosphere, the extinction effect of aerosols
is much greater than that of atmospheric molecules. Therefore, the contribution of extinction by
atmospheric molecules can be neglected. Therefore, the aerosol extinction coefficient can be
deduced from visibility measurements, and the imaginary part of the AERI can be obtained based
on Eq. (4).
Experiments show that the temperature fluctuation satisfies the turbulence "2/3" law(Liu et al.,
2017), and due to small relative changes in pressure and air temperature (unit K) occurring over a
short period, the change in the real part of the AERI has a good linear relationship with the
temperature change, and the fluctuation in the real part of the AERI also satisfies the turbulence
"2/3" law; thus, we can define the structure parameter of temperature, $C_T^2$, and the real part of the
AERISP $C_{n,Re}^2$. Therefore, general scalars can be extended, such as the fluctuation of the imaginary
part of the AERISP and the fluctuation of the atmospheric extinction coefficient. Thus, we can
assume that the imaginary part of the AERI satisfies the turbulence "2/3" law; that is, the structure
function of the imaginary part of the AERI $D_{n,Im}(r)$ ($r$ is the separation) can be defined as
$$D_{n,Im}(r) = \overline{[n_{Im}(\vec{x}) - n_{Im}(\vec{r} + \vec{x})]^2} = C_{n,Im}^2 r^{2/3} \tag{5}$$
where $\vec{x}, \vec{r} + \vec{x}$ are the coordinates of two points in space, $\vec{r}$ is the separation vector, $C_{n,Im}^2$ is the
imaginary part of the AERISP, and the overbar indicates the mean.
Thus, we can introduce the imaginary part of the AERISP $C_{n,Im}^2$, a parameter used to describe
the fluctuation intensity of the imaginary part of the AERI ($C_{n,Im}^2$ should be the structure parameter
for the imaginary part of the AERI, conveniently denoted as the imaginary part of the AERISP).
Correspondingly, we can introduce the structure parameter of the atmospheric extinction
coefficient $C_{\beta_{ext}}^2$ and the structure parameter of the fluctuation of the aerosol mass concentration $C_M^2$.

## 2.2 Two methods of AERI measurement

From the definition of the AERISP in the last part and the relationship between the AERI and the extinction coefficient, it can be seen that the AERISP has an important influence on light propagation in the atmosphere; thus, the AERISP can be estimated from the fluctuations in light propagation intensity and the monitoring of the extinction coefficient. This section describes how to measure the AERISP via two methods.

## 2.2.1 Long-Path Light Propagation Methods

When an approximately collimated light beam in the transparent band of the atmosphere is selected and propagated over a distance, the light intensity at the receiving end fluctuates. The fluctuation in light intensity has two causes: one is the uneven distribution of the real part of the AERI caused by the temperature fluctuation, and the other is the uneven distribution of the imaginary part of the AERI caused by the uneven distribution of aerosol particles. Assuming that the above two causes are not related, they can be decomposed. The power spectral density is usually used to characterize the fluctuation in light intensity. Through spectral analysis, the power spectral density of light intensity fluctuations can be decomposed into the contribution of the imaginary part of the AERISP and the contribution of the real part of the AERISP. The contribution of the inhomogeneous distribution of the imaginary part of the AERISP to the light intensity fluctuation is expressed as the temporal spectrum $W_{lnI,Im}(f)$ (Yuan et al., 2015),

$$W_{\ln I,\mathrm{Im}}(f) = 64\pi^2\eta^2 \int_0^L dx \int_{2\pi f/v}^\infty \Phi_{n,\mathrm{Im}}(\kappa)\cos^2[\frac{\kappa^2 x(L-x)}{2\eta L}][(\kappa v)^2 - (2\pi f)^2]^{-1/2} \cdot$$

$$[\frac{2J_1(\frac{D_r\kappa x}{2L})}{D_r\kappa x/2L}]^2[\frac{2J_1(\frac{D_t\kappa(L-x)}{2L})}{D_t\kappa(L-x)/2L}]^2\kappa d\kappa \qquad (6)$$

where $f$ is the frequency of the log-intensity spectrum, $\eta$ is the wavenumber of the spherical wave ($\eta=2\pi/\lambda$, $\lambda$ is the light wavelength), $x$ is the position of the propagating wave, $L$ is the length of the propagation path, $\kappa$ is the wavenumber of the two-dimensional log-intensity spectrum, and $\Phi_{n,Im}$ is the spectrum of the imaginary parts of the refractive index, where the subscript $n$ denotes the refractive index and the subscript $Im$ denotes the imaginary parts of the refractive index, $D_t$ is the transmitting aperture diameter, $D_r$ is the receiving aperture diameter ($D_t$ and $D_r$ are usually identical for an LAS), $v$ is the transverse wind speed and $J_1$ is the first-order Bessel function. The widely used von Karman spectral form for $\Phi_{n,Im}$ is adopted in this study (Andrews and Phillips, 2005) and can be expressed as follows:

$$\Phi_{n,\mathrm{Im}}(\kappa) = 0.033C_{n,\mathrm{Im}}^2(\kappa^2 + \frac{1}{L_0^2})^{-\frac{11}{6}}e^{-\frac{\kappa^2 l_0^2}{5.92^2}} \qquad (7)$$

Here, $L_0$ is the outer scale of turbulence, and $l_0$ is the inner scale of turbulence.

Substituting Eq. (7) into Eq. (6) and integrating the right-hand side of Eq. (6) yields,

$$W_{\ln I,\mathrm{Im}}(f) = 0.129 C_{n,\mathrm{Im}}^2 \eta^2 L \nu^{5/3} [f^2 + (\frac{\nu}{2\pi L_0})^2]^{-4/3} \tag{8}$$

Using Eq. (8), the imaginary part of the AERISP can be determined based on the shape of the spectrum while being constrained by the low-frequency variance in the light intensity fluctuation from the imaginary part of the AERISP.

To carry out a comparative analysis with the results of the real part of the AERISP, the expression for the power spectral density of the logarithmic light intensity fluctuation $W_{lnI,Re}(f)$ due to the real part of the AERI is also given here as (Clifford, 1971;Nieveen et al., 1998),

$$W_{\ln I,\mathrm{Re}}(f) = 64\pi^2 \eta^2 \int_0^L dx \int_{2\pi f/\nu}^{\infty} \Phi_{n,\mathrm{Re}}(\kappa) \sin^2[\frac{\kappa^2 x(L-x)}{2\eta L}][(\kappa\nu)^2 - (2\pi f)^2]^{-1/2} \cdot$$

$$[\frac{2J_1(\frac{D_r \kappa x}{2L})}{D_r \kappa x/2L}]^2 [\frac{2J_1(\frac{D_t \kappa(L-x)}{2L})}{D_t \kappa(L-x)/2L}]^2 \kappa d\kappa \tag{9}$$

Integrating Eq. (9) yields the fluctuation variance of the log light intensity as

$$\sigma_{\ln I,\mathrm{Re}}^2 = \int_0^{\infty} W_{\ln I,\mathrm{Re}}(f)df = 0.89 C_{n,\mathrm{Re}}^2 L^3 D_t^{-7/6} D_r^{-7/6} \tag{10}$$

The real part of the AERISP is usually calculated using Equ. (10) (Wang et al., 1978).

The calculation steps for the real and imaginary parts of AERISP are as follows: first, power spectrum analysis or correlation analysis of the irradiance fluctuation data are performed; then, the irradiance fluctuation data are decomposed into high-frequency and low-frequency parts; the high-frequency part corresponds to the contribution of the real part of the AERI; and the low-frequency part of the fluctuation corresponds to the contribution of the imaginary part of the AERI; finally, the real part of the AERISP $C_{n,Re}^2$ can be obtained from Eq. (10); and the imaginary part of the AERISP $C_{n,Im}^2$ can be obtained from the low-frequency part of the irradiance fluctuation.

## 2.2.2  Spectral analysis methods for single-point measurements

Aerosol particles experience atmospheric motion, which is consistent with general atmospheric motion characteristics, and the "-5/3" law can be used to characterize fluctuations in aerosol-related properties. Therefore, in the inertial subregion, the extinction coefficient power spectral density is

$$S_{\beta_{ext}}(f) = (2\pi/U)S_{\beta_{ext}}(\kappa) = 0.25 C_{\beta_{ext}}^2 (2\pi/U)^{-2/3} f^{-5/3} \tag{11}$$

The extinction coefficient structure parameter $C_{\beta_{ext}}^2$ can be converted to the imaginary part of the AERISP according to equation (4). The coefficient in Eq. (11) is 0.25(Wyngaard et al., 1971). It has been suggested in the literature that the coefficient for the spectral density should be 0.125(Gibbs and Fedorovich, 2020). The difference between the two coefficients 0.25 and 0.125 is whether the integral of the spectral density is equal to the variance or half of the variance. If the integral of the spectral density is equal to the variance, a coefficient of 0.25 is taken; if the integral of the spectral density is equal to half of the variance, the coefficient is taken as 0.125. According to the spectral density curve, the coefficients are determined within the inertial subregion, and the structure

parameters $C^2_{\beta_{ext}}$ can be obtained. According to the relationship between the extinction coefficient and the imaginary part of the AERI in Eq. (4), the imaginary part of the AERISP can be obtained as $C^2_{n,Im}$.

Similarly, power spectral density profiles with temperature fluctuations that

$$S_T(f) = (2\pi/U)S_T(\kappa) = 0.25C_T^2(2\pi/U)^{-2/3}f^{-5/3} \tag{12}$$

The actual temperature turbulence spectral density profile often takes the form of a von Karman spectrum as

$$S_T(f) = 0.25C_T^2(2\pi/U)^{-2/3}\left(f^2 + (\frac{U}{2\pi L_0})^2\right)^{-5/6} \tag{12'}$$

Based on the relationship between the temperature and the real part of the AERI, we have

$$C^2_{n,Re} = C_T^2/R_{TN}^2 \tag{13}$$

where $R_{TN}$ denotes the coefficient of proportionality between the change in the real part of the AERI and the change in atmospheric temperature (Tatarskii, 1961;Zhou et al., 1991).

$$R_{TN} = \frac{dT}{dn_{Re}} = -1.29\times10^4\times(1+\frac{7.52\times10^{-3}}{\lambda^2})^{-1}\frac{\overline{T}^2}{\overline{P}} \tag{14}$$

where the wavelength λ is in microns, the atmospheric pressure P is in hectopascals, and the temperature T is in K.

The real part of the AERISP can be obtained by fitting the experimental data using Eqs. (12) or (12').

## 2.3 Flux estimation

The method for estimating the AERISP was given in the former sections. The purpose of estimating the AERISP in this paper is to estimate the aerosol flux in the near-surface layer. Here, the method of estimating the aerosol flux based on the AERISP is given first, and then the method of estimating the aerosol vertical transport flux based on the EC technique is introduced.

## 2.3.1 Light propagation method

Experiments have shown that the AERISPs satisfy the theory of surface layer similarity; thus, (Yuan et al., 2019)

$$F_{a\_LAS} = (\frac{C^2_{n,Im}}{C^2_{n,Re}})^{1/2}\frac{R_{MN}}{R_{TN}}u_*\left|T_*\right| \tag{15}$$

where $u_*$ is the friction velocity and $T_*$ is the characteristic potential temperature. These two parameters can be determined from the wind speed and temperature profiles. The real and imaginary parts of the AERISP are determined from LAS measurements. The $R_{MN}$ can be obtained from aerosol mass concentration and visibility measurements ($R_{MN} = M/n_{Im}$, where $M$ is the aerosol mass concentration approximated as $PM_{10}$ and $n_{Im}$ can be determined from visibility measurements)(Yuan et al., 2021), and the $R_{TN}$ can be calculated from the mean air temperature and other measurements using Eq. (14) again.

When turbulence in the surface layer develops, Eq. (15) can be approximated as(Yuan et al., 2019),

$$F_{a\_LAS} = a(\frac{g}{T})^{1/2} R_{TN}^{1/2} (C_{n,\mathrm{Re}}^2)^{1/4} R_{MN} (C_{n,\mathrm{Im}}^2)^{1/2} (z-d) \qquad (16)$$

Here, $a$ is the scale factor with a theoretical value of 0.567 (which needs to be determined by comparative experiments), g is the gravitational acceleration, $z$ is the scintillator height, and $d$ is the zero-plane displacement. Equation (16) does not require measurements of the $u_*$ and $T_*$ data. Generally, the measurement heights are high, and the assumption of developed turbulence in the surface layer is easily met during the day under unstable conditions.

## 2.3.2 Based on single-point eddy covariance

Eddy covariance is a commonly used method for the measurement of Earth air exchange fluxes in the near-surface layer. Using rapid measurements of the vertical wind speed and extinction coefficient to obtain the ups and downs of the vertical wind speed and extinction coefficient, the expression for the vertically transported aerosol flux calculated by the eddy-covariance method with a mean vertical velocity close to zero is given by(Wilczak et al., 2001)

$$F_{a\_EC} = R_{MN} \frac{\lambda}{4\pi} \overline{w' \beta_{ext}'} \qquad (17)$$

The prime' in Eq. (17) denotes fluctuation.

## 2.4 Introduction to the experiment

The experiments were performed on the campus of the University of Science and Technology of China (USTC) in Hefei, Anhui Province, China. The campus of the USTC is located in downtown Hefei. Figure 1a shows part of the Hefei city area, where the red rectangle corresponds to Fig. 1b, the campus of the USTC. The campus is surrounded by four highways, and the two highways in the west and north have more vehicles, especially viaducts in the west. The campus is composed of vegetation, roads and teaching buildings. As shown in Fig. 1b, green vegetation covers most of the campus. The roofs of the school buildings are almost on a plane with the tree canopy and are approximately 17 meters above the ground ($z_H$ =17 m). Thus, the zero-plane displacement was 11.4 m (17 × 0.67=11.4) (Shao et al., 2021;Grimmond and Oke, 1999;Leclerc and Foken, 2014). There are two tall buildings (T and R in Fig. 1b) at the southernmost and northernmost parts of the campus, and the distance between the two buildings is approximately 960 meters. The experiment consists of two parts: one part consists of carrying out the light propagation experiment using a self-developed large aperture scintillator (LAS), and the other part consists of carrying out the measurement using the instruments on the meteorological tower in the middle of the beam (the details of the instruments are listed in Table 1). The transmitting end of the LAS was installed on the 12th floor of the southernmost building (T in Fig. 1b), the receiving end was installed on the 12th floor of the northernmost building (R in Fig. 1b), and the distance of the beam from the ground was approximately 35 meters. The apertures of the transmitting and receiving ends were 250 mm. The sampling frequency of the receiving end was 500 Hz, and a data file was saved every 30 minutes. The height of the meteorological tower is 18 meters above the roof of the teaching building (P in Fig. 1b). The height of the top of the meteorological tower is equal to the height of the beam. The meteorological tower is equipped with 5 layers of wind speed, wind direction, temperature and

humidity measurement sensors. At the top of the tower, there is a radiation quadrature sensor, and at the bottom of the tower, there is a rainfall measurement sensor. In this paper, we use data from the top 18 meters of height of the meteorological tower with sensors installed for conventional meteorological parameters, including temperature, humidity, wind speed, wind direction and radiation. Conventional meteorological data were collected at 1-second intervals, average data were obtained every half hour after data collection, and precipitation data were recorded every half hour. A three-dimensional sonic anemometer thermometer was installed at the top of the tower, and the high-frequency sampling visibility sensor CS120A (Campbell, 2012) was upgraded to obtain 1-Hz visibility (Ren et al. 2020). A three-dimensional sonic anemometer thermometer can obtain a sampling frequency of 10 Hz and is a common instrument used in atmospheric turbulence research; as such, we will not introduce it in depth. To correlate the vertical wind speed with the extinction coefficient to obtain the aerosol flux, the data collected by the sonic anemometer-thermometer at 10 Hz were averaged to obtain 1-Hz data, which were saved in a data file. By doing so, the aerosol flux only contains eddies with a frequency lower than 1 Hz; in other words, any turbulent eddy, whose frequency is higher than 1 Hz, is automatically eliminated. By comparing the T-w correlations calculated from the 10 Hz data and the 1 Hz data, it can be seen that the error due to this high-frequency neglect is less than 5% (details in Appendix).

The time period of the experiment is January 9-23, 2022, a total of 15 days. The winter period was chosen, because it is considered to be typical of this period, with mainly sunny days, weak rainfall, and relatively high pollution in winter.

## 2.5 Data quality control

The quality of the data obtained from field observations needs to be controlled before further processing (Foken and Wichura, 1996). This study involves several types of data, mean variables, cumulative variables, and fluctuating variables. The mean variables included 30-minute averages of temperature, humidity, wind speed, wind direction, and global radiation. Data quality control for mean variables was performed by comparing measurements at different heights or different sites. The same variables with the same trend at different heights and different locations were considered high-quality data. All the measured mean data were determined to be satisfactory. The cumulative variables refer to 30-minute rainfall data. Rainfall data were qualified with reference to relative humidity, total radiation and air temperature. The fluctuating data included 10-Hz ultrasonic anemometer data and 1-Hz visibility data, as well as high-frequency intensity fluctuation data measured by the LAS, the real and imaginary parts of the AERISP, and calculated aerosol fluxes. Quality control consisted mainly of eliminating spikes and replacing missing data.

The reason for the spike points in the light intensity fluctuation data is that the received signal jumps when there are flying birds and other obstructions to the optical signal on the propagation path. This situation is automatically determined by the program. When this occurs, the data for that time period are not processed. The AERISP and aerosol flux data are judged according to (a) three times the standard deviation (SD) from the mean value and (b) three times the standard deviation from the mean of differences between adjacent moment data. To determine the three times the SD from the mean value, the trend is obtained by averaging over a two-hour period, then calculating the difference between the measured value and the trend at each moment, calculating the mean and variance of the difference, and considering a spike point if the difference is outside 3 times the SD. The 3 times the SD of adjacent differences is determined by first calculating the difference between adjacent

moments and then calculating the mean and SD of the difference. Any data that deviates from the
mean by more than 3 times the SD is considered a spike point.
The data judged to be spikes will be supplemented by the average of adjacent moments. Of
course, the data processed according to this method appear to be completely missing for longer time
periods. For such cases, no further methods to realize supplementation are considered in this paper.
There are other errors in measurements made with the LAS due to specific reasons (Moene et al.,
2009); for example, the effect of spectral shape deviations using the von Karman model and
intermittent variations in the properties of this spectrum on the LAS signal are not considered in this
study.
Like for $CO_2$ flux calculations, EC calculations for aerosol flux were performed to obtain
aerosol fluxes, and several data quality control studies were conducted, such as coordinate system
rotations(Wilczak et al., 2001;Yuan et al., 2011), and WPL corrections(Webb et al., 1980).

# 3 Experimental results

In the following, the variation curves of conventional meteorological parameters during the
experimental period, individual examples of AERISPs, a comparison of the two methods for the
results of multiday continuous observations and a comparison of the two methods for the results of
flux measurements are presented to verify the reliability of the means of light propagation
measurements.

## 3.1 General meteorological parameters and extinction coefficients

The variation curves of conventional meteorological parameters during the experiment,
including temperature, humidity, wind speed, wind direction, radiation and precipitation, and
extinction coefficient are shown in Fig. 2, where the extinction coefficient is calculated from the
visibility ($\beta_{ext} = 3.912/V$, $V$ denotes visibility). Seven days during the experiment were sunny, and
four of the remaining eight days had rainfall. The temperatures on sunny days were characterized by
significant daily variations, with a minimum temperature of 0.4°C, and the maximum diurnal
temperature difference could reach more than 9°C. The relative humidity exceeded 80% for only a
few periods during sunny days. The wind speed was generally less than 3 m/s, and there were very
few periods of north wind with a speed greater than 3 m/s. There was no obvious prevailing wind
direction during the experimental period, and only the north wind was equivalent to the other
directions with a slight predominance. The meteorological conditions during the experiment were
similar to those of the local winter season. The extinction coefficient curve with time during the
experiment is given in Fig. 2(g). The pollution gradually increased from the 9th to Jan. 13th and
decreased on the 13th; from the 14th to the 20th, the pollution gradually increased and decreased on
the 20th. The meteorological conditions during the experimental period can be considered typical.

## 3.2 Example results from measurements of the imaginary part of the AERISP

Before carrying out the comparison of the measurement results of the two methods for obtaining the AERISP, the comparison of the measurement results of an individual example is given. The experimental data measured from 2022-01-16 13:00-13:30 will be used here as an example to illustrate the calculation of the AERISP, and the results will be given. This time period is midday on a clear day (shown in Fig. 2e), and both the total radiation and sensible heat fluxes are large, so this time period can be taken as a good typical example.

### 3.2.1 Structure parameters obtained by light propagation

The AERISP is first described using the light propagation method. The sequence of light intensity signals obtained at the receiving end is shown in Fig. 3a. The time duration is 2022-01-16 13:00-13:30, and the sampling frequency is 500 Hz, thus there are 900000 data points in the time series of light intensity fluctuations in Fig. 3a. The curve has both low- and high-frequency fluctuations. Using spectral analysis and correlation analysis, the variance in the low-frequency part of the logarithmic light intensity is 1.08e-4, and the variance in the high-frequency part is 5.06e-4. The solid dots in Fig. 3b are the measured spectral densities of the logarithmic light intensity fluctuations, and the black dashed lines and solid lines represent the results calculated by Eqs. (6) and (9), respectively, and represent the contributions of the imaginary part and the real part. As seen from the power spectral density curves of the logarithmic light intensity fluctuations in Fig. 3b, the high-frequency part and the low-frequency part have different characteristics.

In the logarithmic plot, the low-frequency part is prominent with a much higher spectral density than the high-frequency part. Theoretical analysis revealed that the low-frequency part corresponds to the contribution of the imaginary part of the AERISP. The high-frequency part is flat plus high-frequency attenuation. The high-frequency part corresponds to the contribution of the real part. The part greater than 100 Hz is noise.

Based on the previous theoretical approach, the spectral density fitting for the low-frequency part, while constrained by the low-frequency variance, yields an equivalent refractive index structure parameter of $1.14 \times 10^{-25} m^{-2/3}$. Correspondingly, the structure parameter of the real part of the refractive index, based on the high-frequency variance, is obtained as $2.54 \times 10^{-14} m^{-2/3}$ .

### 3.2.2 Obtaining the imaginary part of the AERISP based on the spectrum

The coefficients of the power spectral density curves are proportional to the refractive index structure parameters from which they can be determined. The extinction coefficient structure parameter can be deduced from the power spectral density of the extinction coefficient fluctuation, and the temperature structure parameter can be deduced from the power spectral density of the temperature fluctuation. The fluctuations in the extinction coefficient (Fig. 4a) and temperature (Fig. 4b) with time for the period 2022-01-16 13:00-13:30 are shown in Fig. 4. As shown in Fig. 4, the

extinction coefficient curve has more noise, while the temperature curve has less noise. On the
temperature fluctuation curve, there are five distinct ramp structures.
Power spectral analysis of the data in Fig. 4 was carried out to obtain the power spectral density
in Fig. 5. From the extinction coefficient power spectral density curve in Fig. 5a, it can be seen that
spectral densities greater than 0.05 Hz exhibit noise, and spectral densities less than 0.05 Hz have
inertial subregions. According to practical analysis, the inertial subregion ranges from 0.002 Hz to
the noise onset frequency. The motion of aerosol particles in the atmosphere conforms to the "-5/3"
law of turbulence. The extinction coefficient structure parameter was obtained by fitting the data in
the inertial subregion using Eq. (11) with a value of $3.9 \times 10^{-11} \text{m}^{-2} m^{-2/3}$, which was then
converted to the structure parameter of the imaginary part of the refractive index of $1.04 \times$
$10^{-25} m^{-2/3}$.
Correspondingly, as seen from the temperature fluctuation power spectrum density curve in Fig.
5b, almost no noise appears, which is mainly due to the small amount of noise in the temperature
signal itself, while the 1 Hz temperature data here are obtained by averaging the data collected at 10
Hz. The temperature structure parameter of $0.0218°\text{C}^2 \text{ m}^{-2/3}$ is obtained by fitting using Eq. (12),
which is converted to a refractive index real part structure parameter of $2.1 \times 10^{-14} m^{-2/3}$.
The imaginary part of the AERISP obtained by using a visibility meter and the real part of the
AERISP obtained by an ultrasonic anemometer are in good agreement with the previous results given
by using optical propagation methods.

## 3.3 Comparison of all the results for the AERISP

The previous section gives an individual example. A comparison of all the data during the
experiment is given below, as shown in Figs 6 and 7.
A comparison of the time series of AERISPs measured by the two methods is given in Fig. 6,
where Fig. 6a shows the time series of the imaginary part of the AERISP and Fig. 6b shows the time
series of the real part of the AERISP. There are large fluctuations in the imaginary part of the AERISP
during the experimental period. This trend is close to that of the aerosol extinction coefficient. Figure
6a shows that there is no obvious daily variation characteristic. The trend agreement of the results
obtained by the two methods is very good. From Fig. 6b, it can be seen that the real part of the
AERISP on sunny days has obvious daily variation characteristics; these characteristics are large
during the day and small at night. The agreement of the results obtained by the two methods is good
during the day (8:00-17:00), and at night, the results obtained by the light propagation method are
greater than those of the large point measurements.
Scatter plots of the results of the measurements of the two methods are given in Fig. 7. Figure
7a shows the scatter plot of the results of the two methods for the imaginary part of the AERISP with
almost the same correlation coefficient $R^2$ for daytime and nighttime, while Fig. 7b shows the scatter
plot of the results of the two methods for the real part of the AERISP with a correlation coefficient
of real $R^2$ of 0.74 for daytime and 0.15 for nighttime. This shows that the correlation coefficients of
the imaginary part of the AERISP obtained by the two methods are almost equal during both daytime
and nighttime, and the correlation coefficient of the real part of the AERISP obtained by the two
methods is smaller at night than during the daytime. This shows that the spatial distribution of aerosol
at night may be more homogeneous than the temperature distribution. The reason for this difference
may be that the temperature distribution in the overlying surface of the campus at night is not uniform,
and weak turbulence does not produce strong mixing, resulting in a nonuniform distribution of the
real part of the AERISP. There are no strong aerosol emission sources on the night-time campus, so
the distribution of the imaginary part of the AERISP behaves more uniformly.

## 3.4 Velocity-extinction coefficient correlation for a single point

To calculate aerosol fluxes using EC techniques, a delayed correlation of the vertical velocity
and extinction coefficient is needed. The delayed correlation curves of the vertical velocity and
extinction coefficient are given in Fig. 8.
The horizontal coordinate of the delay correlation curve in Fig. 8 is the delay time $\tau$, and the
vertical coordinate is the delay correlation. From Fig. 8, it can be seen that at $\tau = -2$ s, the correlation
curve has an obvious extreme value, which is also the minimum value of the delay time for a duration
of 300 s. The minimum value is -5.22e-6 $m^{-1}$. The extreme value of the correlation curve does not
appear at 0 s because there is a distance of approximately 0.20 m between the sensing element of the
visibility meter and that of the ultrasonic anemometer. Here, we present the cases with obvious
extremes, and there are some cases where no obvious extremes appear. In such cases where there are
no significant extremes, the value associated with a delay time of 0 seconds is taken.

## 3.5 Flux

The AERISP was given in the former part, and the aerosol vertical transport flux can be
estimated for the duration of 2022-01-16 13:00-13:30 according to Eq. (16),

$$F_{a\_LAS} = 0.567 * \left(\frac{9.8}{283}\right)^{\frac{1}{2}} * (1.01 \times 10^6)^{\frac{1}{2}} * (2.54 \times 10^{-14})^{\frac{1}{4}} * 6216 * (1.14 \times 10^{-25})^{1/2} * 18 * 10^9 = $$
$$1.60 \ \mu g m^{-2} \ s^{-1} \quad (18)$$

where $a$=0.567, T=283 K, g=9.8 m/s$^2$, $R_{TN} = 1.01 \times 10^6 \ K$, $C_{n,Re}^2 = 2.54 \times 10^{-14} \ m^{-2/3}$, $R_{MN} =$
$6216 \ Kg \cdot m^{-3}$ (Yuan et al., 2015), $C_{n,Im}^2 = 1.14 \times 10^{-25} \ m^{-2/3}$, z=35 m, and d=17 m.
Similarly, the aerosol flux is obtained from the eddy covariance method according to Eq.(17)

$$F_{a\_EC} = -0.522 \times 10^{-6} * 6216 * 10^9 * \frac{0.65 \times 10^{-6}}{4\pi} = -1.67 \ \mu g m^{-2} \ s^{-1} \quad (19)$$

where $\overline{w' \beta_{ext}'} = -0.522 \times 10^{-6} \ s^{-1}$, $R_{MN} = 6216 \ Kg \cdot m^{-3}$, and $\lambda = 0.65 \times 10^{-6} \ m$.
From the previous calculations, we can see that during the half hour from 2022-01-16 13:00-
13:30, the absolute values of the aerosol fluxes obtained by the two methods are very close but of
opposite signs. Since the LAS method based on light propagation cannot determine the direction of
flux transport, only the magnitude of the flux can be determined. This is similar to the fact that the
estimation of surface sensible heat fluxes using an LAS provides information about only the
magnitude but not the direction. There are some judgments for estimating the direction of sensible
heat flux using a LAS, such as those based on sunrise and sunset times and atmospheric stability
(Zhao et al., 2018). Here, a negative flux indicates the deposition of aerosol particles. Because the
experimental site is a campus, there is almost no source of aerosol particle emission in the overlying
surface, which is manifested as a sink of aerosol particles inside the city. Therefore, the direction of
aerosol flux measurements based on the LAS needs to be judged based on the nature of the surface.
The results of aerosol flux calculations throughout the experiment, except for two days of rain,
the 22nd and 23rd days. are given in Fig. 9. Figure 9a shows the absolute values of the aerosol
vertical transport fluxes measured by the two methods based on the imaginary part of the AERISP

and EC methods, and Fig. 9b shows the aerosol vertical transport fluxes with signs for transport direction measured, which correspond to the rectangular-point line in Fig. 9a. The trend of aerosol fluxes obtained by the two methods given in Fig. 9a is consistent with the diurnal variation in aerosol fluxes on sunny days, with larger values of aerosol fluxes at noon. At night, the aerosol flux values are lower. As shown in Fig. 9(a), the absolute value of the aerosol flux obtained by the LAS is greater than that obtained by the EC at noontime on 10-11 Jan, 2022. This is because the imaginary parts of the AERISP obtained by the LAS are larger than those obtained by the EC, as shown in Fig. 6a. Another possible reason is that it was a cloudy day during both the 10th and 11th days, there was a weak rainfall process on the 10th day at 16:00, and the winds on the 10th and 11th days were lighter and had a greater change in direction. The turbulence during noontime on 10-11 is weaker, resulting in an inhomogeneous horizontal distribution and a large difference in measurements between the two methods.

A comparison of Fig. 9a and Fig. 9b,reveals that the aerosol flux is negative at noon on clear days, indicating that the turbulence is strong at noon, which enhances the downward transport of aerosol particles. This study was conducted on a campus with no emission sources, and the downward flux was reasonable; in fact, there was an upward flux measured by the EC method if there were emission sources in the observation area (Ren et al. 2020).

# 4 Conclusion and discussion

To validate the previously developed method of measuring the AERISP and aerosol mass flux, this paper theoretically organizes the concept of the AERISP, introduces two methods for measuring the AERISP and estimating the aerosol vertical transport flux by using the AERISP and EC methods, and carries out field observation experiments in an urban area. The experimental results show that the AERISPs estimated by the two methods are in good agreement, and the aerosol vertical transport fluxes obtained by the two methods based on the AERISP and EC are in good agreement.

According to the experimental results, the imaginary part of the AERISP expresses the intensity of the fluctuation in the attenuation of light during transmission. When the air-transparent band is used, the imaginary part of the AERISP characterizes the intensity of the fluctuation in the extinction coefficient of the aerosol.

The aerosol flux is related to both the fluctuations in aerosol concentration and the intensity of atmospheric turbulence. When there is an aerosol emission source on the overlying surface, the aerosol flux is positive, transporting aerosol particles upwards. When there is no aerosol emission source in the overlying surface, the overall performance is aerosol particle deposition and downwards flux transport. In general, urban green lands are areas of aerosol particle deposition, while ocean and desert surfaces can often be viewed as source areas for aerosols. The large difference in the real part of the AERISP measured by the two methods at night also contributes to the large difference in the aerosol fluxes obtained by the two methods at night.

From the experimental results, we can also see that, as a comparison, this paper also gives results for the temperature refractive index structure parameters, and as shown in Fig. 6, the trends for the structure parameters in the real and imaginary parts of the AERISP are different, indicating that temperature fluctuations and aerosol concentration fluctuations are uncorrelated. The purpose of this paper is to illustrate the physical significance of the imaginary part of the AERISP obtained using the LAS technique and to obtain the aerosol vertical transport flux based on the AERISPs. When inverting the imaginary part of the AERISP using the light propagation principle, it is assumed

that the aerosol concentration fluctuations are not correlated with the temperature fluctuations. This assumption cannot be proven theoretically. From the experimental results, as shown in Fig. 6, the trends of the real and imaginary parts of the AERISP are different, indicating that the temperature fluctuations and the aerosol concentration fluctuations are uncorrelated. This phenomenon shows that the two sources are different and are basically consistent with the actual situation. This also shows that the assumptions of the theory for obtaining the imaginary part of the AERISP are reasonable.

To compare with aerosol transport fluxes obtained based on the AERISPs, this paper uses a delay correlation between the visibility meter and vertical wind speed to obtain aerosol vertical transport fluxes. Currently, a modified visibility meter is utilized to obtain 1 Hz visibility data, after which the extinction coefficient is obtained. The extinction coefficient power spectrum in Fig. 5a shows that there is a large amount of noise in the high-frequency part. The signal-to-noise ratio of the extinction coefficient data is too low compared to the temperature fluctuation or velocity fluctuation, which introduces a large error in the calculation of the aerosol flux. Although the overall trend magnitude agreement of the fluxes obtained by the two methods is good enough to show that the two methods can be corroborated with each other, there are still differences in the details; however, technical methods are required to improve the performance of the instrument and to obtain high-quality aerosol extinction coefficient data to carry out measurements of vertical aerosol transport fluxes based on the EC method at a single point.

**Data availability.** Requests for data that support the findings of this study can be sent to rmyuan@ustc.edu.cn.

**Competing interests.** The authors declare that they have no conflict of interest.

**Author contributions**. Renmin Yuan and Hongsheng Zhang designed experiments and wrote the manuscript; Renmin Yuan, Jiajia Hua, Hao Liu, Xingyu Zhu and Peizhe Wu carried out experiments; Renmin Yuan analyzed experimental results. Jianning Sun revised the manuscript and participated in the discussion.

**Acknowledgements**. This study was supported by the National Natural Science Foundation of China (42075131, 42105076) and the National Key Research and Development Program under grant no. 2022YFC3700701.

# Appendix: Comparison of fluxes between 10 Hz and 1 Hz

To determine the high frequency loss due to the use of 1 Hz data for flux calculations, the T-w covariance was used to perform an analytical comparison between the fluxes obtained by sampling the data at 10 Hz and the fluxes obtained by gaining the data at a frequency of 1 Hz. The data from January 9 and 23, 2022 were processed, and the fluxes corresponding to different sampling frequencies were compared and are shown in Fig.10. There are two ways to obtain 1 Hz data: one is directly obtained at 1 Hz sampling frequency (shown in Fig. 10a), and the other is 1 Hz data obtained by averaging 10 Hz data over 10 data points (shown in Fig. 10b). In comparison, the flux calculated from the 1 Hz data obtained by averaging 10 data points is smaller (slope of 0.97). This indicates a slower response of the instrument. This is the case for the visibility meter, for which a slower

response was used in this study. Based on the linear fit results and the root mean square error (RMSE)
in Fig. 10, the difference in the fluxes between 10 Hz and 1 Hz is less than 5%.
Overall, the error due to the lower sampling frequency of 1 Hz is much smaller than the
difference between the two methods discussed in this study.

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

Table 1 details of all the instruments

| Meteorological elements | Manufacturer type | Sampling frequency (Hz) | Height (m) Above building top |
|---|---|---|---|
| LAS | Self-developed | 500 | 18.0 |
| 3-D sonic anemometer | Campbell CSAT3 | 10 | 18.0 |
| Visibility | Campbell CS120 | 1 | 18.0 |
| Wind speed and direction | 03001 R.M. Young | 1 | 2.0, 4.5, 8.0, 12.0,18.0 |
| Temperature and humidity | Vaisala HMP155A | 1 | 2.0, 4.5, 8.0, 12.0,18.0 |
| Radiation | Kipp&Zonen CNR4 | 1 | 16.0 |
| Precipitation | TE525 Tipping Bucket | 1 | 1.0 |



# **Figures**

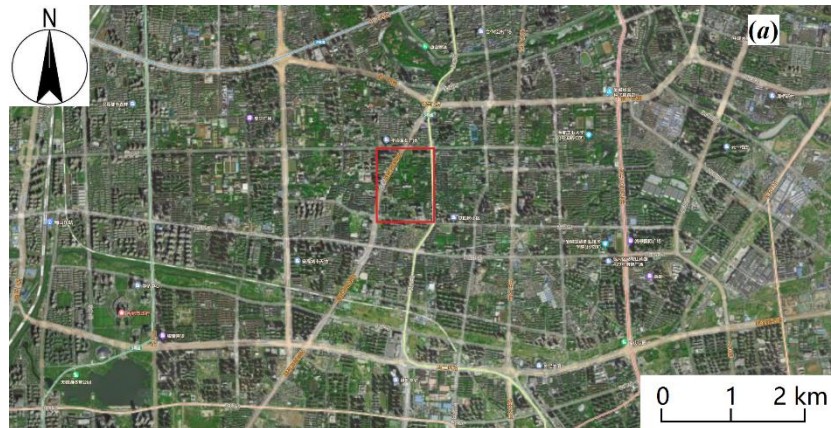

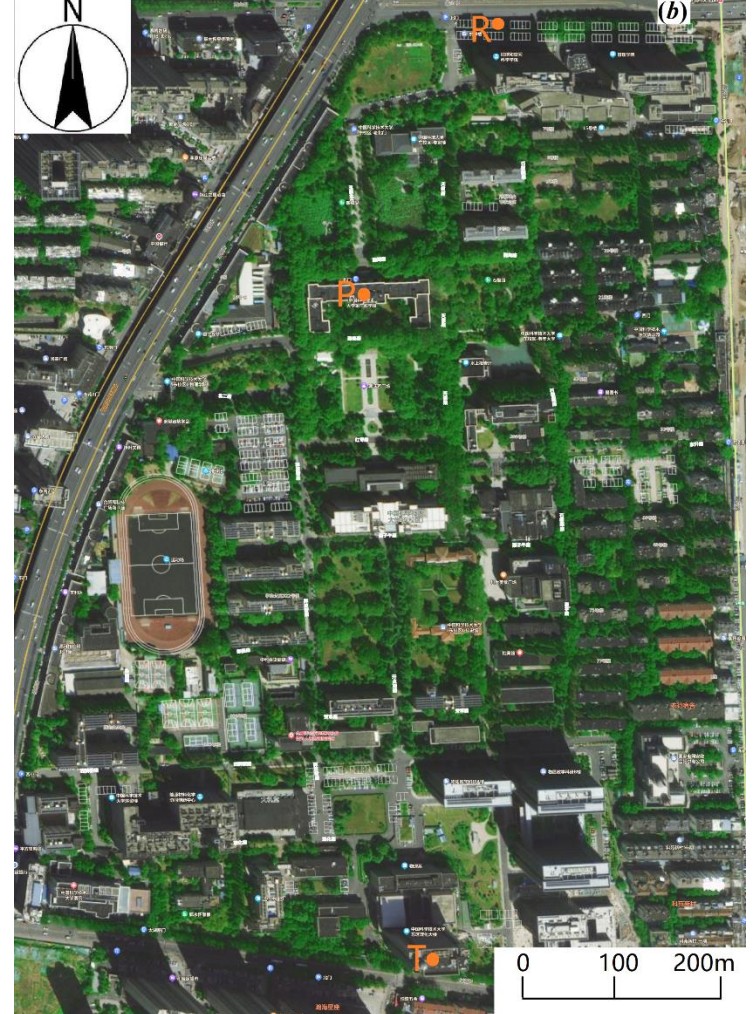

Figure 1. Photographs of the measurement site. (a) Map of Hefei city and (b) expanded view of the
measurement site on the USTC campus, which is marked by the red rectangle in (a). Points T and R in
(b) show the locations of the transmitter and receiver, respectively. Point P in (b) marks the meteorological
tower position. There are four heavy traffic roads surrounding the measurement site. Figure 1a and b @
Baidu are from the following website:
https://map.baidu.com/@13055953.105500832,3719556.851423825,15.3z/maptype%3DB_EART
H_MAP

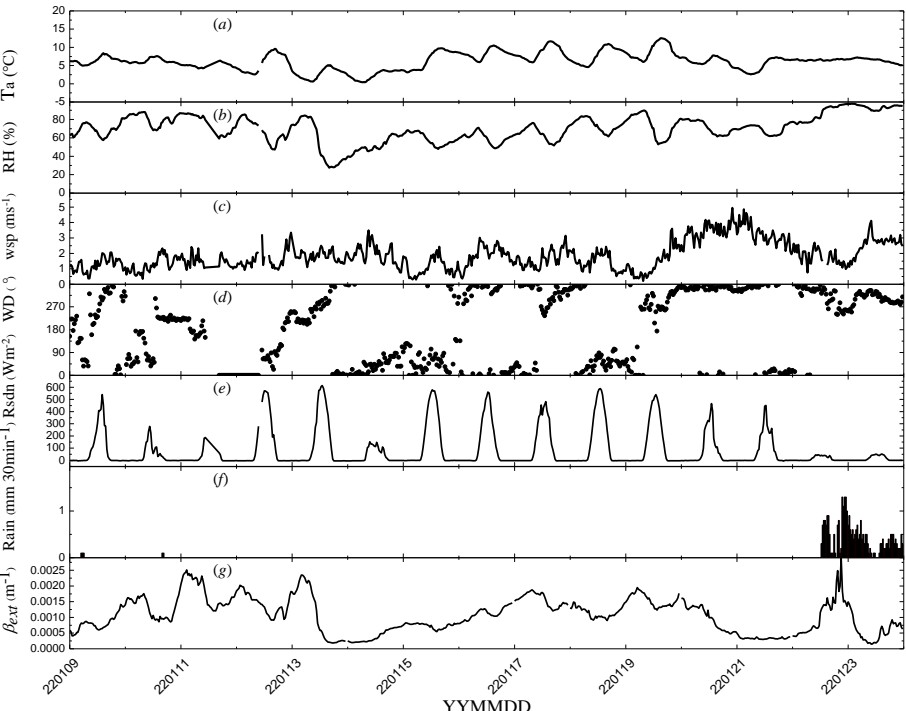



Figure 2. Temporal variations in the (a) air temperature (T), (b) relative humidity (RH), (c) wind
speed (wsp), (d) wind direction (WD), (e) total radiation (Rsdn), (f) precipitation (Rain), and (g)
extinction coefficient ($\beta_{ext}$). The details can be found in the text.

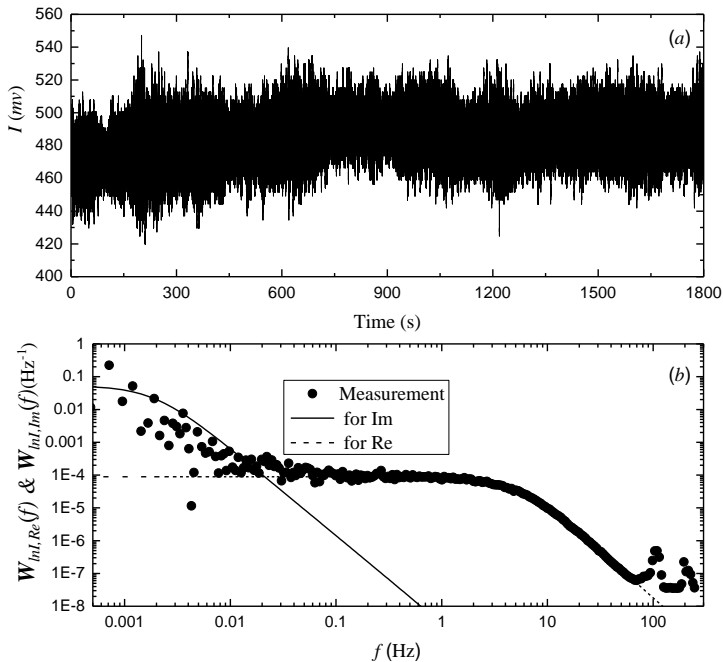


Figure 3 Temporal variations in the light intensity received by the LAS and (b) power spectral
density of the logarithm of the light intensity during 2022-01-16 13:00-13:30.

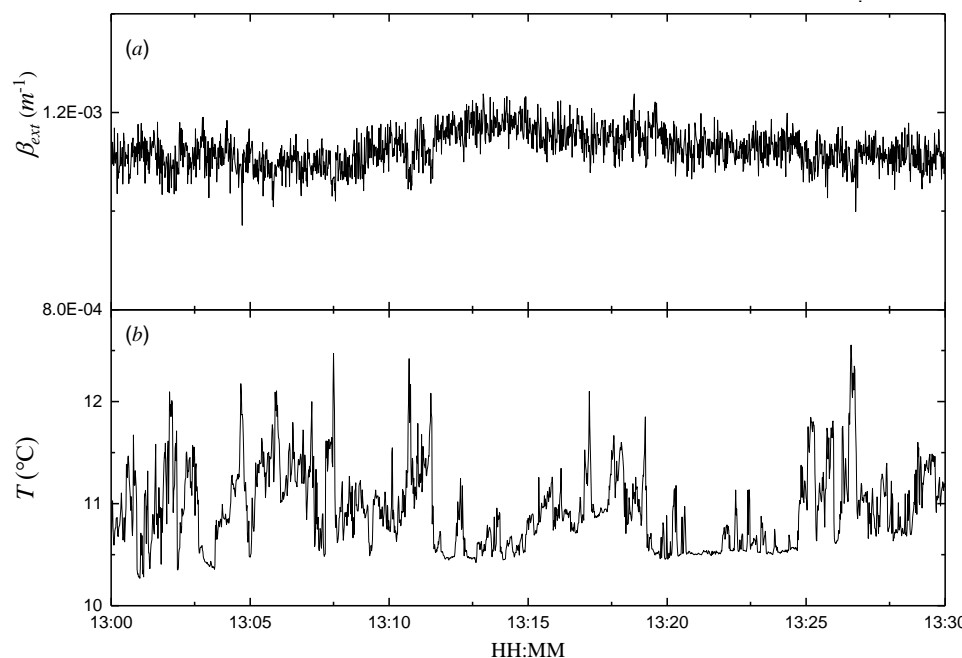


Figure 4 Temporal variations in the (a) extinction coefficient and (b) air temperature during 2022-

718 01-16 13:00-13:30.


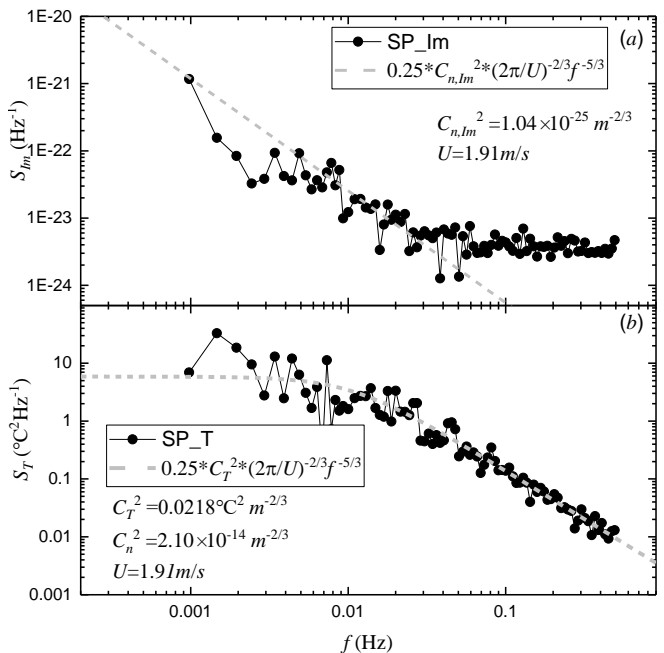


Figure 5. Power spectral density of the (a) extinction coefficient and (b) air temperature during 2022-
01-16 at 13:00-13:30.

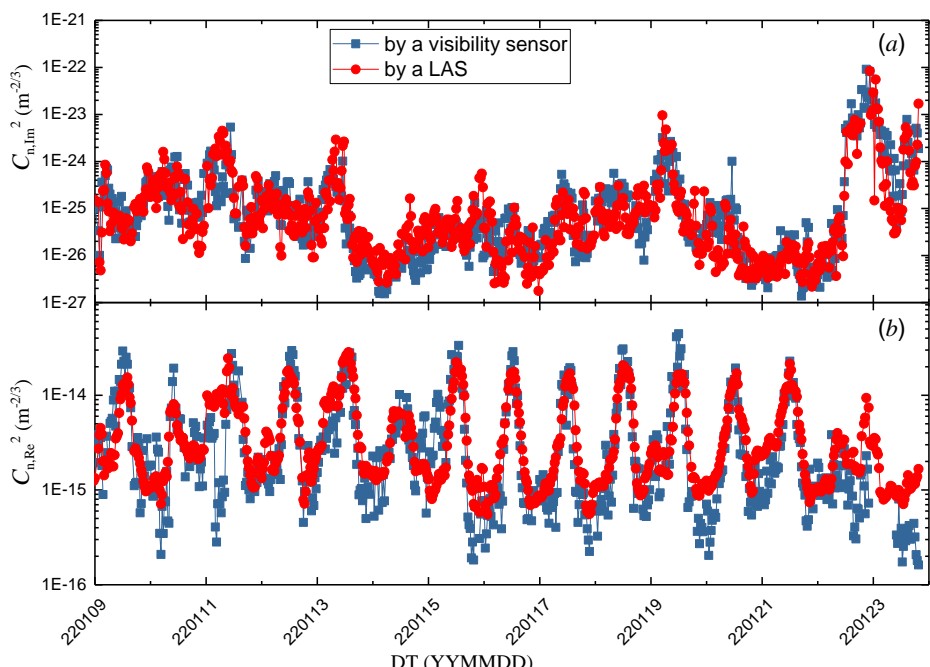


Figure 6. Temporal variations in (a) the imaginary part and (b) real part of the AERISP during 09-
23 Jan. 2022.

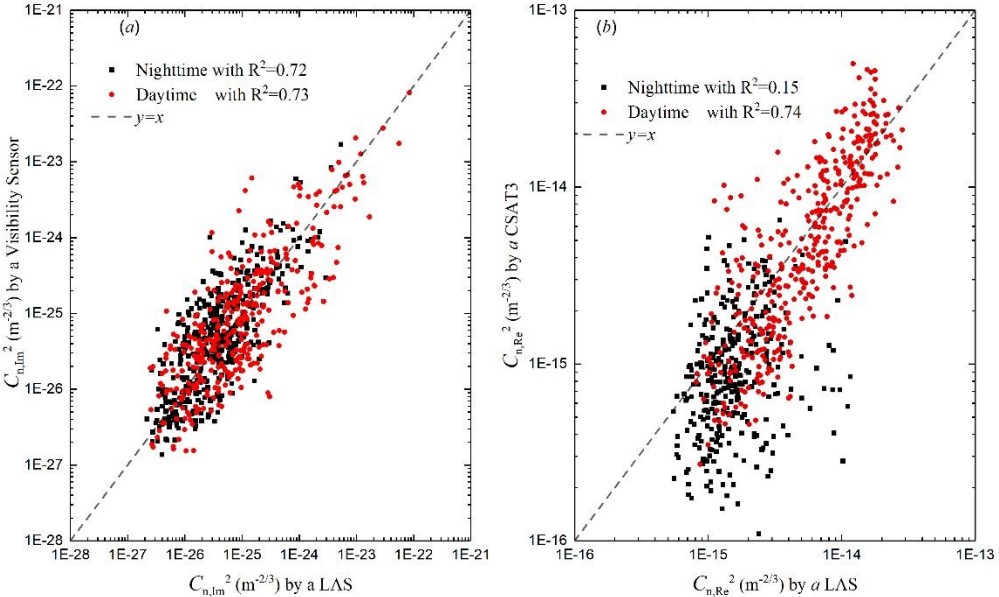



Figure 7. Comparison of (a) the imaginary part and (b) real part of the AERISP during 09-23 Jan. 2022.

729        The red solid circles indicate daytime and the black solid rectangles indicate nighttime.

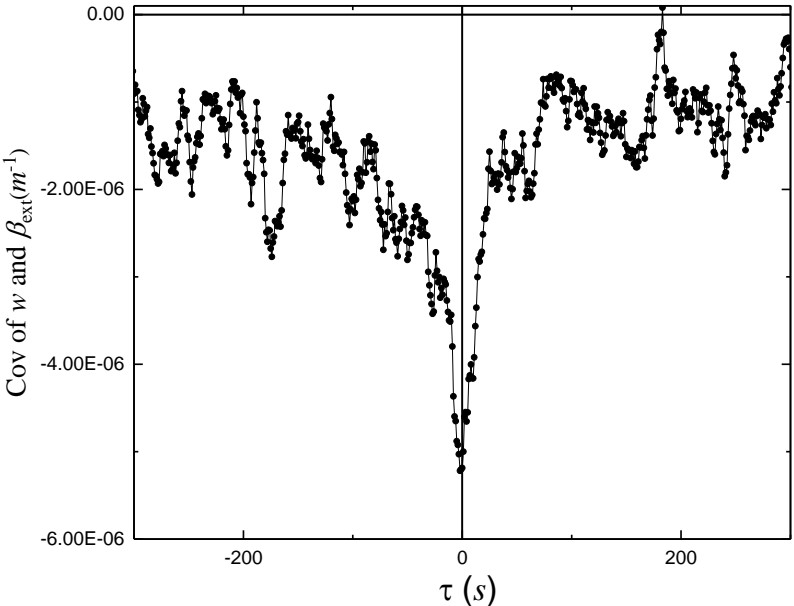


Figure 8. Delay covariance between the extinction coefficient and vertical velocity during 2022-01-
732    16 13:00-13:30.

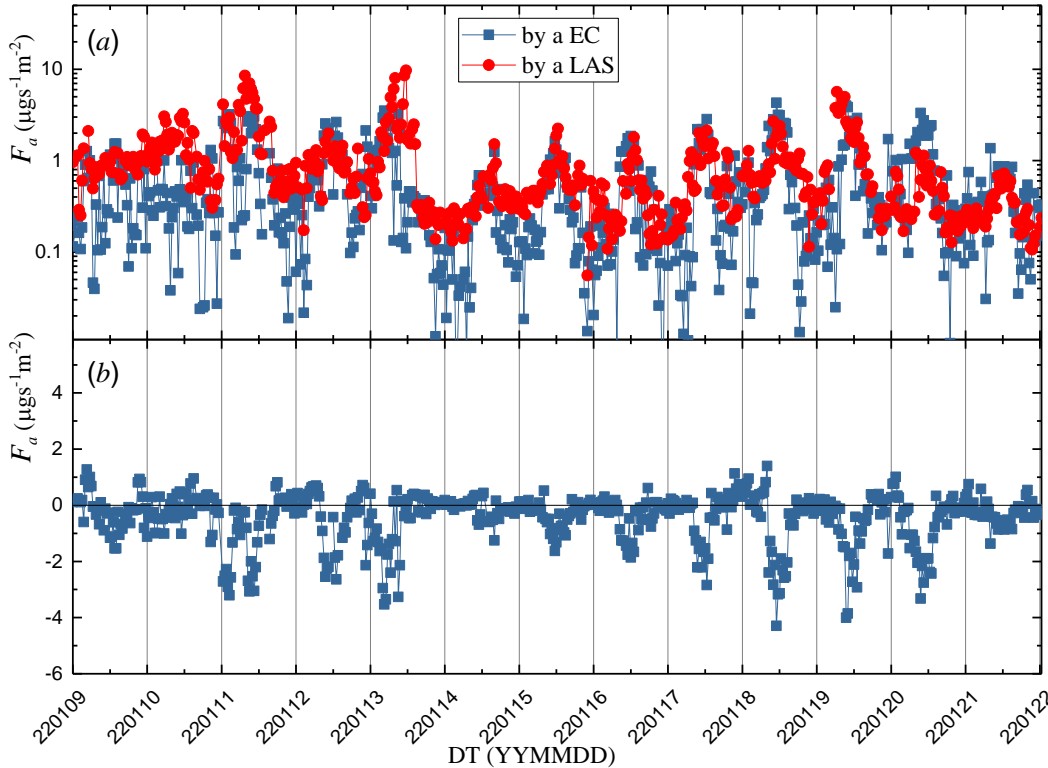


Figure 9. Temporal variations in (a) absolute value of aerosol flux based on the AERISP and EC methods and (b) aerosol flux based on the EC methods during 09-21 Jan. 2022.



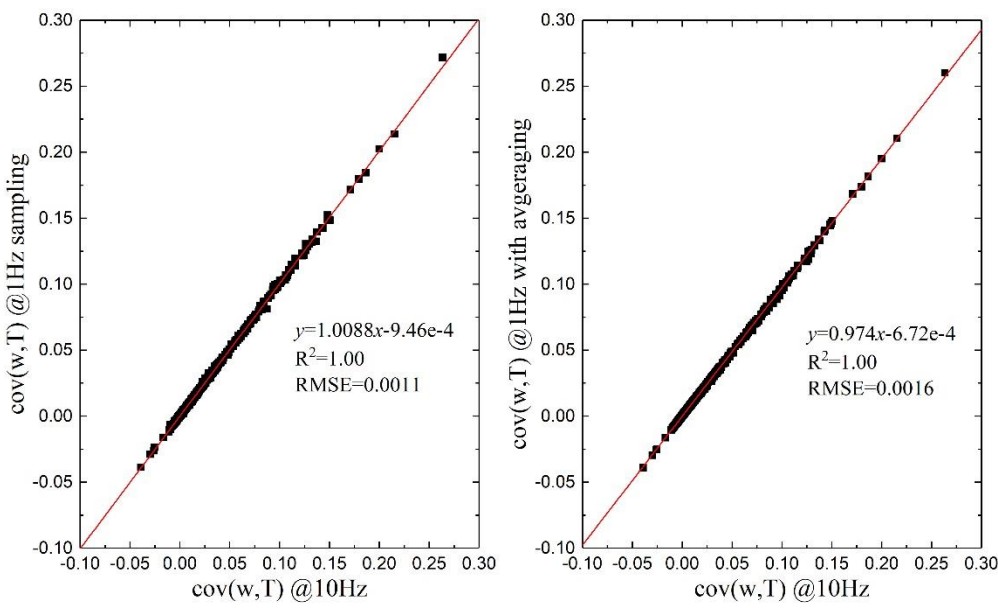


Figure 10 Comparison of covariance of w and T between 10 Hz and 1 Hz, with a 1 Hz sampling rate (a) and 1 Hz data obtained by averaging 10 Hz data over 10 data points (b)
