# Peer review of "Comparison of the imaginary parts of the atmospheric 1 refractive index structure parameter and aerosol flux based 2 on different measurement methods 3"

_EGUsphere, 2023_

## Referee Comment (RC2)

**Title: Comparison of the imaginary part of the atmospheric refractive index structure parameter and aerosol flux based on different measurement methods**
**Author(s): Renmin Yuan et al.**
**MS No.: egusphere-2023-2677**
**MS type: Research article**

The manuscript investigates various aspects related to aerosol vertical transport flux measurements. The authors compare different measurement methods for determining aerosol vertical transport flux, focusing on the validation and mutual agreement among these methods. The study introduces the concept of the atmospheric equivalent refractive index structure parameter (AERISP) and its imaginary part, which is crucial for understanding aerosol concentration fluctuations. The authors propose using the light-propagated large-aperture scintillation to determine the imaginary part of the AERISP. They conduct experiments on the campus of the University of Science and Technology of China to compare AERISPs obtained through different methods and examine the aerosol vertical transport fluxes derived from these measurements.

The experimental results demonstrate good agreement between the imaginary parts of the AERISPs obtained by different methods. The authors also observe consistent trends in aerosol vertical transport fluxes, although some differences exist.

Overall, this manuscript contributes to the improvement of aerosol flux measurements and provides valuable insights into the existing methods. The research questions are well-addressed, and the experimental approach is appropriate. However, this manuscript lacks some of the key information, and a better presentation of this manuscript, in terms of English writing and paper structure, would greatly enhance the manuscript's impact.

Major issues:

1. Line 58-59: Some of the terminologies are a bit confusing. It seems like the atmospheric equivalent refractive index (AERI) and the atmospheric equivalent refractive index structure parameter (AERISP) were not mentioned in Yuan et al. (2015). Is AERISP the same as more commonly used refractive index structure parameter (RISP)? If the authors can elaborate the broadness of these two variables based on the literature, it will certain help the audience understand the whole concept better.

2. Line 65-70: There are a few sub-questions regarding to this part. 1) It is not very clear how the similarity theory is utilized here; 2) Some of the arguments seem strong and would require more references to support. For example, "It is assumed that the aerosol concentration variations follow the same pattern as the scalar

motion", I would like to see authors explain if this argument is an "assumption" or it is observed fact, which should be supported by references.

3. Line 76-92: The purpose of this paragraph is unclear. It shows a lot of details in methodology. It would be much better for the authors rewrite this paragraph focusing on setting up the argument or summarizing research gaps.

4. Line 93: Could the authors provide the definition of the imaginary part of the AERISP?

5. Line 104-107: It is a big part of this work to compute the aerosol vertical transport flux. If I understand correctly, the aerosol flux is calculated by combining 1 Hz visibility and 10 Hz ultrasonic anemometer data, which requires to "downgrade" 10 Hz vertical velocity fluctuation to 1 Hz. By doing so, the aerosol flux only contains the eddy with frequency lower than 1 Hz, in other words, any turbulent eddy, whose frequency is higher than 1 Hz, is automatically eliminated. This brings the argument that if this technique would lose a big part of turbulence information. If the authors can comment on this, that will be great.

6. Line 127-128: Is the equivalent refractive index n_eff = n_equ from Eq. (1)? Please clarify it.

7. Eq. (2): It seems like S(0) is a complex number. If so, the imaginary part of Eq. (1) should be written as $\frac{2\pi}{\eta^3} \int Re[S(0)\frac{dN}{dD}dD]$. Furthermore, the imaginary part of S(0) goes to the real part (i.e., $Re(n_{que}) = n_m - \frac{2\pi}{\eta^3} \int Im[S(0)\frac{dN}{dD}dD]Re(n)$), is that right?

8. Line 136: Please provide more information on the relation between the aerosol extinction coefficient and visibility. At least provide related reference.

9. Line 139-140: The relation between temperature and the real part of the AERI as well as the real part satisfying "2/3" law are not convicting. Consider elaborate more on the arguments.

10. Eq. (5): Please define $D_{n,Im}(r)$

11. Line 163-165: Please consider rephrase this sentence. Five "of"s make this statement hard to follow.

12. Eq. (6): Please define W, D_r, D_j, J_1 and nu. The first integral ($\int_0^L dx$) seems odd. Is this equal to L?

13. Line 262-263: The displacement height is not necessarily equal to the height of the buildings or canopies. A more sophisticated method should be used here to estimate the height of displacement.

14. Please list the details of all the instruments used in this work. For example, the model names of wind speed, direction, temperature, humidity sensors, and sonic anemometer.

15. Line 289: The selection of study periods seems arbitrary. Are there any reasons for this time?
16. Line 421-423: The estimation of AERISP is more accurate during the convective period. I'm wondering if the authors only show the comparison during daytime, how the results would look like.

Minor issues:
1. Line 99-103: "However, the conventional …… and aerosol mass concentration (Ren et al., 2020)". The sentence is way too long to follow. Please consider rephrase it.
2. Line 117-118: "caused by the fluctuation of the refractive index caused by the fluctuation of temperature" Please rephrase.
3. Figure 1: Please add scales and orientation in (a). Please consider change the color of letters and add scales and orientation as well. If authors can mark the distances between each of these three points, it would be a lot nicer.
4. Line 418: "Fig. 7a" should be Fig. 7b.
5. Line 645: "Figure 8(a,)"
6. Line 649: "and (b) the imaginary part and (b) aerosol flux" Please rephrase.

---

## Author Comment (AC1)

Authors' reply to the reviewer's comments:

Dear Anonymous Referees,

Thank you for your careful review of the manuscript. We read the reviewer's comments carefully, and have responded to and taken all of the comments into consideration and revised the manuscript accordingly. My detailed responses are as follows:

**Comments from Anonymous Referee #1:**

**This work tried to compare the atmospheric equivalent refractive index structure parameter measured by two different methods and then compare the aerosol mass vertical transport fluxes obtained by different methods. The paper is interesting and generally well-written. However, there are several questions need to be answered before its acceptance.**

**Major concerns:**

**In Section 3.3, in Fig.7, the correlation coefficient of the real part of the AERISP is smaller, and the reason is that the difference in the real part of the AERISP obtained by the two methods at night is larger. I wonder this reason can also explain the big discrepancy of the absolute value of aerosol flux obtained by the LAS and EC during nighttime in Fig.9(a)? And also in Fig.9(a), why the absolute value of aerosol flux obtained by the LAS is larger than EC at noontime on 10-11,12 Jan, 2022?**

**Answer:** This can also explain the large discrepancy in the absolute values of aerosol fluxes obtained by the LAS and EC at night in Fig.9(a) because the calculation of aerosol flux needs the value of the real part of the AERISP, which is given in Eq. (16).

We thank the reviewers for their careful observation. The reason why the absolute value of the aerosol flux obtained by the LAS was greater than that obtained by the EC at noontime on 10-11 Jan, 2022, was that the imaginary parts of the AERISP obtained by the LAS were greater than that obtained by the EC, shown in Fig. 6a. A further possible explanation is that both days 10th and 11th were cloudy, there was a weak rainfall process on the 10th at 16:00, and the winds on the 10th and 11th were lighter and had a greater change in direction. The turbulence during noontime on 10-11 is weaker, resulting in an inhomogeneous horizontal distribution and a large difference in measurements between the two methods. On the 12th, sunny, and at midday on the 12th, the measurements taken by the two methods were relatively close to each other.

Please see Lines 519-526.

"As shown in Fig. 9(a), the absolute value of the aerosol flux obtained by the LAS is greater than that obtained by the EC at noontime on 10-11 Jan, 2022. This is because the imaginary parts of the AERISP obtained by the LAS are larger than those obtained by the EC, as shown in Fig. 6a. Another possible reason is that it was a cloudy day during both the 10th and 11th days, there was a weak rainfall process on the 10th day at 16:00, and the winds on the 10th and 11th days were lighter and had a greater change in direction. The turbulence during noontime on 10-11 is weaker, resulting in an inhomogeneous horizontal distribution and a large difference in measurements between the two methods." (Paragraphs in blue are copied from the revised manuscript)

**In Section 3.5, how can you get the values of $C 2n$, and , $C 2n, I m$ ?**

Answer: When a light beam propagates in the atmosphere, irradiance fluctuations at the end of an LAS can be attributed to two causes. One is the inhomogeneous distribution of the atmospheric gas refractive index due to temperature fluctuations, which refracts and diffracts the beam, and the other is the existence of aerosol particles in the atmosphere, which scatter and absorb the beam. The first causes high-frequency fluctuations in light intensity, and the second causes low-frequency fluctuations in light intensity. By analyzing the light intensity fluctuation spectrum, it is possible to separate high-frequency fluctuations from low-frequency fluctuations.

The calculation steps for the real and imaginary parts of the AERISP are as follows: first, power spectrum analysis or correlation analysis of the irradiance fluctuation data are performed; then, the irradiance fluctuation data are decomposed into high-frequency and low-frequency parts; the high-frequency part corresponds to the contribution of the real part of the AERI, and the low-frequency part of the fluctuation corresponds to the contribution of imaginary part of the AERI; finally, the real part of the AERISP $C_{n,Re}^2$ can be obtained from the high-frequency part of the irradiance fluctuation, and the imaginary of the AERISP $C_{n,Im}^2$ can be obtained from the low-frequency part of the irradiance fluctuation.

Please see Lines 226-232.

"The calculation steps for the real and imaginary parts of AERISP are as follows: first, power spectrum analysis or correlation analysis of the irradiance fluctuation data are performed; then, the irradiance fluctuation data are decomposed into high-frequency and low-frequency parts; the high-frequency part corresponds to the contribution of the real part of the AERI; and the low-frequency part of the fluctuation corresponds to the contribution of the imaginary part of the AERI; finally, the real part of the AERISP $C_{n,Re}^2$ can be obtained from Eq. (10); and the imaginary part of the AERISP $C_{n,Im}^2$ can be obtained from the low-frequency part of the irradiance fluctuation. "

**Specific comments:**

**Page 3, line 95-96, the meaning of "However, the direct and rapid measurement of the atmospheric extinction coefficient is difficult to realize" is not very clear**

At present, the existing online instruments for measuring atmospheric extinction coefficient include visibility meters, LIDAR devices and nephelometers, etc. The sampling period of these instruments is 1 minute at the shortest, which makes it difficult to perform rapid measurements of the atmospheric extinction coefficient.

This paragraph has been deleted in the revised version.

**Line 279, "wind direction radiation"?**

Thank you, we modified this text.

**Line 645: Figure 8 (a,) -> Figure 8**

Thank you, we modified this text.

**Line 649: "and (b) the imaginary part and (b) aerosol flux"?**

Thank you, we modified this text.

We also modified some typo errors.

Finally, the authors thank the two referees for their constructive comments,which helped us to improve the clarity and quality of the manuscript greatly. All the comments are answered and the modifications are provided in the revised manuscript. We sincerely hope our answers can relieve doubts and provide a better description of our work.

---

## Author Comment (AC2)

Authors' reply to the reviewer's comments:

Dear Anonymous Referees,

Thank you for your careful review of the manuscript. We read the reviewer's comments carefully, and have responded to and taken all of the comments into consideration and revised the manuscript accordingly. My detailed responses are as follows:

**Comments from Anonymous Referee #2:**

**Title: Comparison of the imaginary part of the atmospheric refractive index structure parameter and aerosol flux based on different measurement methods Author(s): Renmin Yuan et al.**

**MS No.: egusphere-2023-2677**

**MS type: Research article**

**The manuscript investigates various aspects related to aerosol vertical transport flux measurements. The authors compare different measurement methods for determining aerosol vertical transport flux, focusing on the validation and mutual agreement among these methods. The study introduces the concept of the atmospheric equivalent refractive index structure parameter (AERISP) and its imaginary part, which is crucial for understanding aerosol concentration fluctuations. The authors propose using the light-propagated large-aperture scintillation to determine the imaginary part of the AERISP. They conduct experiments on the campus of the University of Science and Technology of China to compare AERISPs obtained through different methods and examine the aerosol vertical transport fluxes derived from these measurements.**

**The experimental results demonstrate good agreement between the imaginary parts of the AERISPs obtained by different methods. The authors also observe consistent trends in aerosol vertical transport fluxes, although some differences exist.**

**Overall, this manuscript contributes to the improvement of aerosol flux measurements and provides valuable insights into the existing methods. The research questions are well-addressed, and the experimental approach is appropriate. However, this manuscript lacks some of the key information, and a better presentation of this manuscript, in terms of English writing and paper structure, would greatly enhance the manuscript's impact.**

**Major issues:**

1. **Line 58-59: Some of the terminologies are a bit confusing. It seems like the atmospheric equivalent refractive index (AERI) and the atmospheric equivalent refractive index structure parameter (AERISP) were not mentioned in Yuan et al. (2015). Is AERISP the same as more commonly used refractive index structure parameter (RISP)? If the authors can elaborate the broadness of these two variables based on the literature, it will certain help the audience understand the whole concept better.**

Answer: The AERISP in the present manuscript is the ARISP in Yuan et al. (2015).

The more commonly used refractive index structure function $D_n$ was defined (Tatarskii,),

$$D_n(\vec{x}, \vec{r} + \vec{x}) = \overline{[n(\vec{x}) - n(\vec{r} + \vec{x})]^2} \tag{1}$$

where $n$ is the refractive index, $\vec{x}, \vec{r} + \vec{x}$ are the coordinates of two points in space, and the overbar indicates the mean. For locally homogeneous isotropic turbulence, we have,

$$D_n(\vec{x}, \vec{r} + \vec{x}) = D_n(r) \tag{2}$$

For separation $r$ in the inertial range of scales, $l_0 \ll r \ll L_0$ ($l_0$ for the turbulence inner scale and $L_0$ for the turbulence outer scale),

$$D_n(r) = C_n^2 r^{2/3} \tag{3}$$

$C_n^2$ in Eq. (3) is the more commonly used refractive index structure parameter (RISP).

The separation $r$ is usually less than one meter, so the more commonly used refractive index structure parameter (RISP) $C_n^2$ is for one position. The atmospheric refractive index structure constant (RISP) is conventionally referred to for gases in the atmosphere. In contrast, our measurements here were conducted by long-path light propagation, and the transmitter end was approximately 960m from the receiver end. The AERI and the AERISP are introduced by treating the aerosol and the gas in the propagation path as a whole, this process results in an equivalent medium.

In a physical sense, the AERISP is the same as the RISP, which is why the RISP was initially referred to as the AERISP (Yuan, et al., 2015). However, by definition, the more commonly used RISP is for the atmosphere at a certain position in space, while the AERISP is for the equivalent medium over long ranges (typically hundreds of meters and kilometers). It was later felt that the AERISP would be more appropriate. This means that the RISP in the Yuan et al. (2015) is the same as the AERISP mentioned above.

Please see Lines 58-62.

"Previously, Yuan et al. (2016) introduced the concepts of the atmospheric equivalent refractive index (AERI) and the atmospheric equivalent refractive index structure parameter (AERISP). The term AERISP corresponds to the equivalent medium containing air and aerosol particles, relative to the commonly used atmospheric refractive index structure parameter (RISP)

obtained from single-point measurements (Wyngaard et al., 1971)." (Paragraphs in blue are copied from the revised manuscript)

2. **Line 65-70: There are a few sub-questions regarding to this part. 1) It is not very clear how the similarity theory is utilized here; 2) Some of the arguments seem strong and would require more references to support. For example, "It is assumed that the aerosol concentration variations follow the same pattern as the scalar motion", I would like to see authors explain if this argument is an "assumption" or it is observed fact, which should be supported by references.**

Answer:

For the temperature structure parameter $C_T^2$, which is assumed to follow similarity theory (Wyngaard et al., 1971),

$$\frac{C_T^2(z-d)^{2/3}}{T_*} = g_3\left(\frac{z-d}{L}\right) \tag{4}$$

where $z$ is the measurement height, $d$ is the zero-displacement height, $L$ is the Monin–Obukhov (MO) length and $T_*$ is the surface-layer characteristic temperature, $g_3$ which has the form

$$\begin{cases} g_3 = 4.9(1 - 7\frac{z-d}{L})^{-2/3} & \frac{z-d}{L} \leq 0 \\ g_3 = 4.9(1 + 2.75\frac{z-d}{L}) & \frac{z-d}{L} \geq 0 \end{cases} \tag{5}$$

For $-\frac{z-d}{L} \gg 1$, and $g_3 \cong \frac{4}{3}4.9(-\frac{z-d}{L})^{-2/3}$, the surface heat flux $Q_s$ can be obtained as follows

$$Q_s = a(C_T^2)^{\frac{3}{4}}\left(\frac{g}{\overline{T}}\right)^{\frac{1}{2}}(z-d) \tag{6}$$

where $a$ is a constant, $g$ is the acceleration of gravity, and $\overline{T}$ is the mean temperature.

Experiments have shown that aerosol particles follow the same theory of locally homogeneous isotropic turbulence as air molecules (Martensson et al., 2006;Vogt et al., 2011a), that is, the fluctuation in the particle concentration follows the '-5/3' power law under unstable atmospheric stratification, and the concentration-velocity cospectra for particle number flux follow the '-4/3' power law, similar to the temperature-velocity cospectra (Kaimal et al., 1972). Therefore, the distribution of small particles can be regarded as a passive conservative quantity, similar to the temperature field. Then, it can be assumed that the aerosol mass concentration fluctuation also follows the locally homogeneous isotropic turbulence theory, that is, the aerosol mass concentration (denoted as $M$) structure function ($D_M(\vec{r})$) in a locally homogeneous isotropic field follows the "2/3 law" $D_M(\vec{r}) = C_M^2 r^{2/3}$, thus the aerosol mass concentration structure parameter $C_M^2$ can be defined (Yuan et al., 2016).

The structure function of the aerosol mass concentration fluctuation ($C_M^2$) can also be assumed to follow similarity theory (Yuan et al., 2016;Yuan et al., 2019),

$$\frac{C_M^2 (z-d)^{2/3}}{M_*} = g_3\left(\frac{z-d}{L}\right) \tag{7}$$

where characteristic parameter $M_*$ for the aerosol mass concentration, which is similar to $T_*$. Then, the formulation for the aerosol mass flux $F_a$ can be given for $-\frac{z-d}{L} \gg 1$,

$$F_a = a\left(\frac{g}{\bar{T}}\right)^{\frac{1}{2}} (C_T^2)^{\frac{1}{4}} (C_M^2)^{\frac{1}{2}} (z-d) \tag{8}$$

Using the relationship between the temperature-real part of the AERI and the aerosol mass concentration-imaginary part of the AERI, the temperature structure parameter and aerosol mass concentration structure parameter are obtained from the real part of the AERISP and imaginary part of the AERISP,

$$C_T^2 = R_{TN}^2 C_{n,\text{Re}}^2 \tag{9}$$

$$C_M^2 = R_{MN}^2 C_{n,\text{Im}}^2 \tag{10}$$

where $R_{TN}$ and $R_{MN}$ are the coefficients for temperature-real part of the AERI and the aerosol mass concentration-imaginary part of the AERI respectively.

Please see Lines 68-81.

"Experiments have shown that aerosol particles follow the same theory of locally homogeneous isotropic turbulence as air molecules(Martensson et al., 2006;Vogt et al., 2011b;Ren et al., 2020); that is, the fluctuation in the particle concentration follows the '-5/3' power law under unstable atmospheric stratification, and the concentration-velocity cospectra for particle number flux follow the '-4/3' power law, similar to the temperature-velocity cospectra (Kaimal et al., 1972). Therefore, the distribution of small particles can be regarded as a passive conservative quantity, similar to the temperature field.

Then, it can be assumed that the aerosol mass concentration fluctuation also follows the locally homogeneous isotropic turbulence theory; thus the aerosol mass concentration structure parameter can be defined (Yuan et al., 2016). Based on the fact that the temperature structure function satisfies the surface layer similarity theory and thus the surface layer sensible heat flux is obtained from the temperature structure parameter (Wyngaard et al., 1971), it is analogous to that, based on the fact that the aerosol mass concentration structure parameter satisfies the surface similarity theory, the surface layer aerosol mass flux is obtained from the aerosol mass concentration structure parameter (Yuan et al., 2016;Yuan et al., 2019)."

3. **Line 76-92: The purpose of this paragraph is unclear. It shows a lot of details in methodology. It would be much better for the authors rewrite this paragraph focusing on setting up the argument or summarizing research gaps.**

Answer:

This paragraph focuses on our intention to use another method of measuring aerosol fluxes, i.e., particle number concentration fluxes. Several studies have combined particle number concentration measurements with vertical wind speed measurements in eddy-covariance (EC) systems for aerosol number concentration flux measurements. We followed this approach and conducted several measurements in 2019 and 2020 to compare the results of the particle number concentration flux measurements with those obtained based on the optical transmission method. However, the results obtained by the two methods are in poor agreement. Our analysis suggested that the fluxes measured by the two methods are not the same. The aerosol number concentration flux must be combined with parameters such as the particle size distribution, complex refractive index of aerosol particles and aerosol particle density; these parameters also need to be sampled in real time, and then aerosol mass flux can be calculated for comparison with one based on the light-propagated large-aperture scintillation principle. It is almost impossible to obtain the particle size distribution and complex refractive index of aerosol particles in real time, so the two methods are not consistent in many cases. This also illustrates the complexity of aerosol particles.

We have modified the document to make the meaning more fluent. Please see Lines 94-114.

"At present, in addition to the previously mentioned measurements of the AERISP based on the principle of long-range light propagation and the similarity theory of the surface layer to obtain the vertical transport flux of aerosol mass in the surface layer, several studies utilize eddy covariance (EC) techniques with fluctuations in aerosol particle number concentration and fluctuation in vertical wind speed to obtain the flux of the number concentration of aerosol particles. Such measurements are carried out in many places (Gordon et al., 2011;Vogt et al., 2011a;Ripamonti et al., 2013). Measurements have revealed quantitative relationships for urban aerosol fluxes among urban vehicle emissions and meteorological conditions (Jarvi et al., 2009), and characteristics of sea salt transport, and aerosol properties (Nemitz et al., 2009). We followed this approach and conducted several measurements in 2019 and 2020 to measure aerosol particle number concentration fluxes using an eddy-correlation system combining a fast-response particle counter with an ultrasonic anemometer with a response frequency of up to 10 Hz, and to calculate aerosol mass concentration fluxes by simultaneously measuring aerosol particle size distribution, mass concentrations, forward scattering coefficients, extinction coefficients, and other parameters. For half of the experimental period, the trends of the measurements of the two methods were the same, while the other periods differed more (unpublished experimental results). The main reason may be the very weak extinction of aerosol particles at scales much smaller than the working wavelength. Second, the aerosol number concentration flux must be combined with parameters such as particle size distribution, complex refractive index of aerosol particles and aerosol particle density, which also need to be sampled in real time. This also illustrates the complexity of aerosol particles."

4. **Line 93: Could the authors provide the definition of the imaginary part of the AERISP?**

Answer:

The AERI consists of the real part ($n_{Re}$) and the imaginary part ($n_{Im}$), $n_{equ} = n_{Re} + i \cdot n_{Im}$, thus, the AERI structure function

$$D_{n_{equ}}(\vec{x}, \vec{r} + \vec{x}) = \overline{[n_{Re}(\vec{x}) - n_{Re}(\vec{r} + \vec{x})]^2} + i^2 \cdot \overline{[n_{Im}(\vec{x}) - n_{Im}(\vec{r} + \vec{x})]^2} + \\ 2i\overline{[n_{Re}(\vec{x}) - n_{Re}(\vec{r} + \vec{x})][n_{Im}(\vec{x}) - n_{Im}(\vec{r} + \vec{x})]} \tag{11}$$

The structure functions for the real part and imagine part are defined as

$$\begin{cases} D_{n_{Re}}(\vec{x}, \vec{r} + \vec{x}) = \overline{[n_{Re}(\vec{x}) - n_{Re}(\vec{r} + \vec{x})]^2} \\ D_{n_{Im}}(\vec{x}, \vec{r} + \vec{x}) = \overline{[n_{Im}(\vec{x}) - n_{Im}(\vec{r} + \vec{x})]^2} \end{cases} \tag{12}$$

For a locally homogeneous isotropic turbulence,

$$D_{n_{Re}}(r) = C_{n,Re}^2 r^{2/3}, \quad D_{n_{Im}}(r) = C_{n,Im}^2 r^{2/3} \tag{13}$$

It is assumed that the fluctuations in the real and imaginary parts of the refractive index are not correlated (Filho et al., 1983), thus,

$$D_{n_{equ}}(r) = (C_{n,Re}^2 + i^2 C_{n,Im}^2) r^{2/3} \tag{14}$$

This equation defines the structure parameters of the real part and the imaginary part of the AERI.

$C_{n,Im}^2$ should be the structure parameter for the imaginary part of the AERI, and is conveniently denoted as the imaginary part of the AERISP.

Please see Lines 177-179.

"Thus, we can introduce the imaginary part of the AERISP C_(n,Im)^2, a parameter used to describe the fluctuation intensity of the imaginary part of the AERI (C_(n,Im)^2 should be the structure parameter for the imaginary part of the AERI, conveniently denoted as the imaginary part of the AERISP)."

5. **Line 104-107: It is a big part of this work to compute the aerosol vertical transport flux. If I understand correctly, the aerosol flux is calculated by combining 1 Hz visibility and 10 Hz ultrasonic anemometer data, which requires to "downgrade" 10 Hz vertical velocity fluctuation to 1 Hz. By doing so, the aerosol flux only contains the eddy with frequency lower than 1 Hz, in other words, any turbulent eddy, whose frequency is higher than 1 Hz, is automatically eliminated. This brings the argument that if this technique would lose a big part of turbulence information. If the authors can comment on this, that will be great.**

Answer:

To determine the high frequency loss due to the use of 1 Hz data for flux calculations, the T-w covariance was used to perform an analytical comparison between the fluxes obtained by sampling the data at 10 Hz and the fluxes obtained by gaining the data at a frequency of 1 Hz. The

data from January 9 and 23, 2022 were processed, and the fluxes corresponding to different sampling frequencies were compared and are shown in Fig.10.There are two ways to obtain 1 Hz data, one is directly obtained at 1 Hz sampling frequency (shown in Fig. 1a), and the other is 1 Hz data obtained by averaging 10 Hz data over 10 data points (shown in Fig. 1b). In comparison, the flux calculated from the 1 Hz data obtained by averaging 10 data is smaller (slope of 0.97). This indicates a slower response of the instrument. This is the case for the visibility meter used in this study, but the error is less than 5%.

Overall, the error due to the lower sampling frequency of 1 Hz is small, much smaller than the difference between the two methods discussed in this study.

[Figure]

Figure 1  Comparison of covariance of w and T between 10 Hz and 1 Hz, with 1Hz sampling rate (a) and 1 Hz data obtained by averaging 10 Hz data over 10 data points (b)

Please see Lines 333-335, and Lines 588-602 in the Appendix.

"By comparing the T-w correlations calculated from the 10 Hz data and the 1 Hz data, it can be seen that the error due to this high-frequency neglect is less than 5% (details in Appendix)."

6. **Line 127-128: Is the equivalent refractive index n_eff = n_equ from Eq. (1)? Please clarify it.**

Answer: the n_eff is n_equ. The symbol n_eff in Line 148 has been modified to n_equ.

7. **Eq. (2): It seems like S(0) is a complex number. If so, the imaginary part of Eq. (1)should be written as $\frac{2\pi}{\eta^3}\int_0^\infty Re[S(0)\frac{dN}{dD}dD]$. Furthermore, the imaginary part of S(0) goes to the real part (i.e., $Re(n_{equ}) = n_m - \frac{2\pi}{\eta^3}\int_0^\infty Im\left[S(0)\frac{dN}{dD}dD\right]Re(n)$), is that right?**

Answer: Yes, You are correct, and S(0) is a complex number.

Because $\frac{dN}{dD}dD$ is a real number, $Re[S(0)\frac{dN}{dD}dD] = Re[S(0)]\frac{dN}{dD}dD$.

Yes, the imaginary part of S(0) goes to the real part, but the imaginary part of S(0) is much less than $n_m$.

8. **Line 136: Please provide more information on the relation between the aerosol extinction coefficient and visibility. At least provide related reference.**

Answer:

Visibility is usually referred to as the maximum horizontal distance through the atmosphere that objects can be seen by the unaided human eye (Alec Bennett, 2014). The minimum brightness contrast value that the human eye can distinguish from a large enough object at a distance is 0.02. Then, based on the dependence of the reduction in contrast on atmospheric absorption and scattering, the following relationship between visibility $V$ and extinction coefficient $\beta_{ext}$ ($V=3.912/\beta_{ext}$) can be obtained (Middleton, 1957; Charlson_1969). Thus, $\beta_{ext}$ in the relationship ($V=3.912/\beta_{ext}$) represents the extinction by all compositions in the air, e.g. absorption and scattering of aerosols and atmospheric molecular extinction. In other words, the visibility-based extinction coefficient is the sum of the extinction coefficient from aerosol absorption and scattering and atmospheric molecular extinction coefficient. However, in the urban atmosphere, the extinction effect of aerosols is much greater than that of atmospheric molecules. Therefore, the contribution of extinction by atmospheric molecules can be neglected.

Middleton, W.E.K., Vision through the Atmosphere. in: Bartels, J. (Ed.), Encyclopedia of physics, University of Toronto Press, Toronto (1957), p. 1054.

Charlson, R. J.: Atmospheric visibility related to aerosol mass concentration - a review, Environmental Science & Technology, 3, 913-918, 10.1021/es60033a002, 1969.

Please see Lines 156-160.

"Due to the dependence of the reduction in contrast on atmospheric absorption and scattering, the following relationship between visibility $V$ and extinction coefficient $\beta_{ext}$ can be obtained: $V=3.912/\beta_{ext}$ (Middleton, 1957;Charlson, 1969). Thus, $\beta_{ext}$ in the relationship ($V=3.912/\beta_{ext}$) represents the extinction by all compositions in the air, e.g., absorption and scattering of aerosols and atmospheric molecular extinction. In other words, the visibility-based extinction coefficient is sum of the extinction coefficient from aerosol absorption and scattering and the atmospheric molecular extinction coefficient."

9. **Line 139-140: The relation between temperature and the real part of the AERI as well as the real part satisfying "2/3" law are not convicting. Consider elaborate more on the arguments.**

Answer:

If the gas composition and aerosol particles in the atmosphere are treated as a whole, AERI ($n_{equ}$) is expressed as (Barrera et al., 2007; Calhoun et al., 2010; van de Hulst, 1957):

$$n_{equ} = n_m + i\frac{2\pi}{\eta^3}\int_0^\infty I_m[S(0)]\frac{dN}{dD}dD \tag{16}$$

The right part of Eq. (4) consists of two items: the first ($n_m$) denotes the air refractive index, which represents the contribution of gas molecules, and the second item denotes the contributions of aerosol particle scattering and absorption. In Eq. (4), $k$ is the working light wavenumber and $i$ represents the imaginary number. S(0) is the amplitude function of the forward-scattering wave of aerosol particles (0 in the arc is the scattering angle) (van de Hulst, 1957), and depends on aerosol particle scattering and absorption.

Let $n_{Re}$ and $n_{Im}$ represent the real and imaginary parts of the AERI, respectively i.e., $n_{equ}=n_{Re}+in_{Im},$; then,

$$n_{Re} = R_e(n_m) - \frac{2\pi}{\eta^3}\int_0^\infty I_m[S(0)]\frac{dN}{dD}dD \tag{17}$$

$$n_{Im} = I_m(n_m) + \frac{2\pi}{\eta^3}\int_0^\infty R_e[S(0)]\frac{dN}{dD}dD \tag{18}$$

If the atmospheric transparent band is selected as the working wavelength, the contribution of molecular scattering to the extinction coefficient is very small relative to the contribution of aerosols; therefore, the contribution of molecular scattering is neglected here. The refractive indices of air molecules in Eq. (4) then have only a real part. Moreover, the second term on the right-hand side of Eq. (5) is much smaller than the first, thus

$$n_{Re} = n_m \tag{19}$$

$$n_{Im} = \frac{2\pi}{\eta^3}\int_0^\infty R_e[S(0)]\frac{dN}{dD}dD \tag{20}$$

Based on the following equation (Tatarskii, 1963; Zhou et al, 1991),

$$n_{Re} - 1 = 77.6 \times 10^{-6} \times (1 + \frac{7.52\times10^{-3}}{\lambda^2})\frac{P}{T} \tag{21}$$

where P is the atmospheric pressure (hPa), T is the air temperature (K) and $\lambda$ is the work wave length (μm). We have,

$$dn_{Re} = 77.6 \times 10^{-6} \times (1 + \frac{7.52\times10^{-3}}{\lambda^2})\frac{\bar{p}}{\bar{T}}(\frac{dp}{\bar{p}} - \frac{dT}{\bar{T}}) \tag{22}$$

Because $\frac{p'}{\bar{p}} \ll \frac{T'}{\bar{T}}$, so the last Eq. (10) can be written.

$$dn_{Re} = -77.6 \times 10^{-6} \times (1 + \frac{7.52\times10^{-3}}{\lambda^2})\frac{\bar{p}}{\bar{T}^2}dT \tag{23}$$

It follows that the working wavelength is constant, usually accompanied by small relative changes in pressure and air temperature (unit K) over a short period, and that the change in the real part of the refractive index of the air has a good linear relationship with the temperature change,

thus suggesting that the relation between temperature and the real part of the AERI as well as the real part satisfies the "2/3" law.

Zhou, X., Tao, S., and Yao, K.: Advaned atmospheric physics, Meteorological Publishing House, Beijing, 1991.

Tatarskii, V. I.: Wave Propagation in a Turbulent Medium, McGraw-Hill Book Company Inc., New York, 1961.

van de Hulst, H. C.: Light Scattering by Small Particles, John Wiley & Sons, Inc., New York, 1957.

Calhoun, W. R., Maeta, H., Combs, A., Bali, L. M., and Bali, S.: Measurement of the refractive index of highly turbid media, Opt. Lett., 35, 1224-1226, 2010.

Barrera, R. G., Reyes-Coronado, A., and Garcia-Valenzuela, A.: Nonlocal nature of the electrodynamic response of colloidal systems, Phys. Rev. B, 75, 184202, 10.1103/PhysRevB.75.184202, 2007.

Please see Lines 165-169.

"Experiments show that the temperature fluctuation satisfies the turbulence "2/3" law(Liu et al., 2017), and due to small relative changes in pressure and air temperature (unit K) occurring over a short period, the change in the real part of the AERI has a good linear relationship with the temperature change, and the fluctuation in the real part of the AERI also satisfies the turbulence "2/3" law; thus, we can define the structure parameter of temperature, $C_T^2$, and the real part of the AERISP $C_{n,Re}^2$."

10. **Eq. (5): Please define $D_{n_{Im}}(r)$.**

Answer:

We add definitions for $D_{n_{Im}}(r)$, and $n, \vec{x}, \vec{r} + \vec{x}$.

Please see Lines 171-176.

"Thus, we can assume that the imaginary part of the AERI satisfies the turbulence "2/3" law; that is, the structure function of the imaginary part of the AERI $D_{n,Im}(r)$ ($r$ is the separation) can be defined as

$$D_{n,Im}(r) = \overline{[n_{Im}(\vec{x}) - n_{Im}(\vec{r} + \vec{x})]^2} = C_{n,Im}^2 r^{2/3} \qquad (5)$$

where $\vec{x}, \vec{r} + \vec{x}$ are the coordinates of two points in space, $\vec{r}$ is the separation vector, $C_{n,Im}^2$ is the imaginary part of the AERISP, and the overbar indicates the mean."

11. **Line 163-165: Please consider rephrase this sentence. Five "of"s make this statement hard to follow.**

Answer:

The power spectral density is usually used to characterize the fluctuations in light intensity. Through spectral analysis, the power spectral density of light intensity fluctuations can be decomposed into the contribution of the imaginary part of the AERISP and the contribution of the

real part of the AERISP. The contribution of the inhomogeneous distribution of the imaginary part of the AERISP to the light intensity fluctuation is expressed as Eq.(6) in the revised version.

Please see Lines 195-200.

"Through spectral analysis, the power spectral density of light intensity fluctuations can be decomposed into the contribution of the imaginary part of the AERISP and the contribution of the real part of the AERISP. The contribution of the inhomogeneous distribution of the imaginary part of the AERISP to the light intensity fluctuation is expressed as the temporal spectrum $W_{lnI,Im}(f)$ (Yuan et al., 2015)"

12. **Eq. (6): Please define W, D_r, D_j, J_1 and nu. The first integral($\int_0^L dx$)seems odd. Is this equal to L?**

Answer:

We thank the reviewer for your careful review and for pointing out our omissions. These symbols were defined in the revised version. Please see Lines 208-210.

"$D_t$ is the transmitting aperture diameter, $D_r$ is the receiving aperture diameter ($D_t$ and $D_r$ are usually identical for an LAS), $v$ is the transverse wind speed and $J_1$ is the first-order Bessel function"

The first integral is $\int_0^L dx = L$, but Eq.(6) is a double integral with x inside the second integral.

13. **Line 262-263: The displacement height is not necessarily equal to the height of the buildings or canopies. A more sophisticated method should be used here to estimate the height of displacement.**

Answer: Yes, the displacement height is not necessarily equal to the height of the buildings or canopies.

There is an error here. The zero-plane displacement of the observation site had been calculated previously (Shao et al. 2021).

According to the methods of Grimmond and Oke (Grimmond and Oke. 1999), the plan aerial fraction and frontal area index were taken as influencing factors to calculate the zero-plane displacement, which was close to the value by the simple method in Leclerc and Foken( 2014), namely, $d=0.67*z_H=11.4$; thus $d$ was taken as 11.4 m.

Please see Lines 303-305.

"The roofs of the school buildings are almost on a plane with the tree canopy and are approximately 17 meters above the ground ($z_H =17$ m). Thus, the zero-plane displacement was 11.4 m ($17 \times 0.67=11.4$) (Shao et al., 2021;Grimmond and Oke, 1999;Leclerc and Foken, 2014)."

Shao B, Yuan R, Liu H, Qiao B, Wang Z, et al. 2021. Research on Turbulence Characteristics in Urban Rough Sublayer-Taking a Site in Hefei as an Example. Journal of atmospheric and environmental optics 16:307

Grimmond, C. S. B., and Oke, T. R.: Aerodynamic Properties of Urban Areas Derived from Analysis of Surface Form, Journal of Applied meteorolgy, 38, 1262-1292, 1999.

14. **Please list the details of all the instruments used in this work. For example, the model names of wind speed, direction, temperature, humidity sensors, and sonic anemometer.**

Please see Table 1 on Line 697.

15. **Line 289: The selection of study periods seems arbitrary. Are there any reasons for this time?**

The winter period was chosen, because it is considered to be typical of this period, with mainly sunny days, weak rainfall, and relatively high pollution in winter.

Please see Lines 336-338.

"The time period of the experiment is January 9-23, 2022, a total of 15 days. The winter period was chosen, because it is considered to be typical of this period, with mainly sunny days, weak rainfall, and relatively high pollution in winter."

16. **Line 421-423: The estimation of AERISP is more accurate during the convective period. I'm wondering if the authors only show the comparison during daytime, how the results would look like.**

Answer:

The AERISPs obtained during the day and night were compared, as shown in Figure 2. As shown in Fig. 2, the agreement of the data of the imaginary part of the AERISP measured by different methods during the daytime (8:00-17:00) is approximately the same as that during the nighttime, while the agreement of the data of the real part of the AERISP measured by different methods during the daytime is much better than that during the nighttime.

[Figure]

Figure 2 Comparison of the AERISP from 09-23 Jan. 2022.

The red solid circles indicate daytime and the black solid rectangles indicate nighttime.

Please see Lines 466-473.

"Figure 7a shows the scatter plot of the results of the two methods for the imaginary part of the AERISP with almost the same correlation coefficient $R^2$ for daytime and nighttime, while Fig. 7b shows the scatter plot of the results of the two methods for the real part of the AERISP with a correlation coefficient of real $R^2$ of 0.74 for daytime and 0.15 for nighttime. This shows that the correlation coefficients of the imaginary part of the AERISP obtained by the two methods are almost equal during both daytime and nighttime, and the correlation coefficient of the real part of the AERISP obtained by the two methods is smaller at night than during the daytime."

**Minor issues:**

1. **Line 99-103: "However, the conventional …… and aerosol mass concentration (Ren et al., 2020)". The sentence is way too long to follow. Please consider rephrase it.**

We modified the paragraph. Please see Lines 115-125.

2. **Line 117-118: "caused by the fluctuation of the refractive index caused by the fluctuation of temperature" Please rephrase.**

We modified the paragraph. Please see Lines 134-138.

3. **Figure 1: Please add scales and orientation in (a). Please consider change the color of letters and add scales and orientation as well. If authors can mark the distances between each of these three points, it would be a lot nicer.**

We modified Figure 1.

4. **Line 418: "Fig. 7a" should be Fig. 7b.**

Thank you, we modified this text.

5. **Line 645: "Figure 8(a,)"**

Thank you, we modified this text.

6. **Line 649: "and (b) the imaginary part and (b) aerosol flux" Please rephrase.**
Thank you, we modified this text.

We also modified some typo errors.

Finally, the authors thank the two referees for their constructive comments,which helped us to improve the clarity and quality of the manuscript greatly. All the comments are answered and

the modifications are provided in the revised manuscript. We sincerely hope our answers can relieve doubts and provide a better description of our work.